# Easy2Hard-Bench: Standardized Difficulty Labels for Profiling LLM Performance and Generalization

Mucong Ding[*][†]      Chenghao Deng[*][†]      Jocelyn Choo[†]      Zichu Wu[△]
Aakriti Agrawal[†]      Avi Schwarzschild[□]      Tianyi Zhou[†]      Tom Goldstein[†]
John Langford[α]      Anima Anandkumar[⋈]      Furong Huang[†]

{mcding,dengch16,furongh}@umd.edu

## Abstract

While generalization over tasks from easy to hard is crucial to profile language models (LLMs), the datasets with fine-grained difficulty annotations for each problem across a broad range of complexity are still missing. Aiming to address this limitation, we present Easy2Hard-Bench, a consistently formatted collection of 6 benchmark datasets spanning various domains, such as mathematics and programming problems, chess puzzles, and reasoning questions. Each problem within these datasets is annotated with numerical difficulty scores. To systematically estimate problem difficulties, we collect abundant performance data on attempts to each problem by humans in the real world or LLMs on the prominent leaderboard. Leveraging the rich performance data, we apply well-established difficulty ranking systems, such as Item Response Theory (IRT) and Glicko-2 models, to uniformly assign numerical difficulty scores to problems. Moreover, datasets in Easy2Hard-Bench distinguish themselves from previous collections by a higher proportion of challenging problems. Through extensive experiments with six state-of-the-art LLMs, we provide a comprehensive analysis of their performance and generalization capabilities across varying levels of difficulty, with the aim of inspiring future research in LLM generalization. The datasets are available at https://huggingface.co/datasets/furonghuang-lab/Easy2Hard-Bench.

## 1 Introduction

The development and evaluation of Large Language Models (LLMs) depend crucially on their ability to generalize across a broad spectrum of tasks, ranging from basic to complex problem-solving scenarios. However, among the current prevalent benchmarks, only a select few include problems with annotated difficulty levels. These annotations are typically presented as categorical values [37, 43, 4, 24] or through pairwise comparisons [52], neither of which provide a detailed portrayal of the difficulty distribution within the dataset. Such granularity is essential for effectively benchmarking and enhancing the adaptability and training approaches of LLMs. To address this gap, there is a pressing need for a benchmark that provides numerical difficulty estimations for problems across various domains.

In previous datasets, difficulty estimation has typically been based on domain-specific characteristics, such as language similarity in linguistic reasoning [5] and equations in mathematical problems [52], or a few human validators' opinions on each problem [37]. Relying solely on these features makes it challenging to rate problem difficulty uniformly across a continuum and restricts the evaluation

---

[*]Equal contribution. [†]University of Maryland, [△]University of Waterloo, [□]Carnegie Mellon University, [⋈]California Institute of Technology, [α]Microsoft.

38th Conference on Neural Information Processing Systems (NeurIPS 2024) Track on Datasets and Benchmarks.

Table 1: Overview of Easy2hard-Bench. Easy2hard-Bench consists of six datasets: E2H-AMC, E2H-Codeforces, and E2H-Lichess are newly created with difficulties estimated from human statistics, while E2H-GSM8K, E2H-ARC, and E2H-Winogrande are existing datasets with continuous difficulty estimations from thousands of LLMs on the *Open LLM Leaderboard* [5]. These datasets cover diverse domains such as math, coding, puzzles, and reasoning, which have well-recognized difficulty concepts. Item Response Theory (IRT) [31, 38] and Glicko-2 (advanced Elo rating) [19] are used for difficulty estimation for each sample, both providing difficulty uncertainties.

| | Topic | Source | Statistics Used to Infer Difficulty | Source Type | Estimation Method |
|---|---|---|---|---|---|
| **E2H-AMC** | Math Competitions | AMC, AIME, HMMT | Item difficulties | Human | IRT |
| **E2H-Codeforces** | Competitive Programming | Codeforces [45] | Submission status & contestant ratings | Human | Glicko-2 |
| **E2H-Lichess** | Chess Puzzles | Lichess [46] | Player ratings & puzzle ratings | Human | Glicko-2 |
| **E2H-GSM8K** | Math Word Problems | [12] | Sample-wise evaluation results of thousands of LLMs on *Open LLM Leaderboard* [5] | LLMs | IRT |
| **E2H-ARC** | Natural Science QA | [11] | | | |
| **E2H-Winograde** | Commonsense Reasoning | [40] | | | |

process to domains that are easily interpretable by humans. A more effective alternative could be the quantitative analysis of interactions between problems and examinees, allowing for a continuous-valued difficulty rating. This method does not depend on the domain nature; it posits that problems deemed more difficult can only be solved by examinees with sufficient capability, whereas easier problems are solvable by a broader range of examinees. By collecting extensive performance data from a large pool of humans or models for each problem, a numerical difficulty score can be accurately assigned using statistical models to regress this data.

Motivated by this concept, we introduce the Easy2Hard-Bench, a benchmark comprising six datasets, each with an estimated difficulty rating for every problem. This benchmark is distinguished by the following features:

- **Rich Domain Diversity:** The Easy2Hard-Bench spans six distinct domains, including mathematics, programming, chess, and various reasoning tasks. These diverse tasks encompass a broad spectrum of prevalent cognitive challenges for LLMs.
- **Continuous Difficulty Rating:** The difficulty of problems is estimated using continuous values, employing advanced statistical models such as Glicko-2 [19] and Item Response Theory (IRT) [38, 34]. This methodology utilizes abundant real-world performance results from humans and leaderboard data from LLMs, providing a clearer insight into the difficulty structure of each dataset.
- **Distribution of Difficulty:** As visualized in Figure 2, the problems within each domain cover a wide range of difficulties. Including more problems with higher difficulty could further reveal the limits of current LLM capabilities.
- **Unified Data Format:** The difficulty ratings for problems across all domains are presented in a consistent format, making all datasets within the benchmark user-friendly for LLM workflows.

**Summary of Contributions:**

1. We present an innovative dataset to employ a refined, continuous valuation of difficulty. This refined methodology for labeling difficulty serves the current needs and sets a precedent for future datasets, particularly as AI systems advance and tasks become increasingly complex.
2. The large scale of the dataset, both in terms of breadth and depth, enables a granular assessment of LLM capabilities across a spectrum of complexity in various domains.
3. To ensure the difficulty labels reflect genuine cognitive hurdles, we incorporate human-verified difficulty assessments, adding a layer of realism and relevance.
4. By introducing rigorous, reasoning-focused challenge tasks with the dataset, the dataset not only tests but also seeks to expand the current boundaries of what LLMs can achieve.
5. Our proposed methodology for assigning continuous-valued difficulty levels will be invaluable for future dataset curation efforts, particularly as AI models become more capable and the tasks designed to challenge them grow more complex. This approach ensures that datasets can evolve in tandem with advancements in AI capabilities, maintaining relevance and challenge over time.

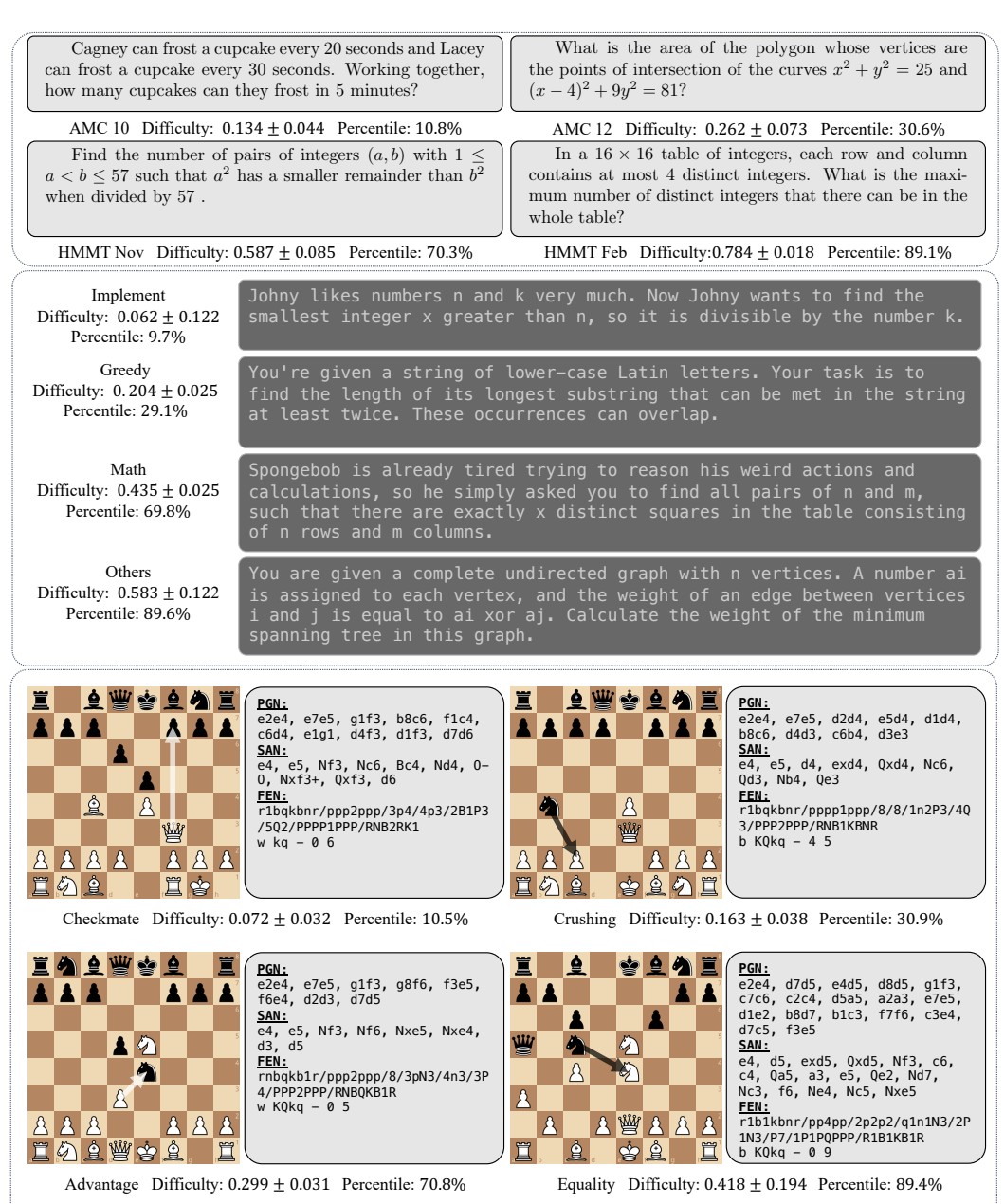

Figure 1: Example problems at different difficulty levels. We present problems from **E2H-AMC**, **E2H-Codeforces**, and **E2H-Lichess** datasets, illustrating varying difficulty levels within each domain. Higher estimated difficulties correspond to more complex problems, as verified by human studies.

We encourage the AI community to engage with the Easy2Hard-Bench, envisioning it as a catalyst for pioneering studies, innovative training methodologies, and the development of AI systems that truly interact with the complexities of the human cognitive landscape.

## 2 Easy2Hard-Benchmarking Suite

**Dataset Overview and Statistics.** The Easy2Hard-Bench (see Table 1) offers a diverse combination of challenges designed to rigorously assess the capabilities of large language models (LLMs) across varying complexity levels. This benchmark suite comprises six diverse datasets, meticulously curated

Table 2: Statistics of three newly curated datasets in Easy2Hard-Bench, see Appendix I for the others.

| | Sizes | | Average Difficulty | Avg. # of Tokens | | # of Tags | Categories |
|---|---|---|---|---|---|---|---|
| | Train | Eval | | Problem | Solution | | |
| **E2H-AMC** | 1,000 | 2,975 | .437 ± .244 | 99.6 ± 34.6 | 31.3 ± 93.8 | 20 | *AMC8, AMC10, AMC12, AIME, HMMT-Nov, HMMT-Feb* |
| **E2H-Codeforces** | 3,663 | 4,000 | .331 ± .190 | 509.2 ± 7.5 | 325.4 ± 231.7 | 37 | *Greedy, Implement, Math, DP, Others* |
| **E2H-Lichess** | 71,763 | 5,000 | .307 ± .156 | 1607.8 ± 253.3 | 7.5 ± 0.3 | 45 | *Checkmate, Advantage, Crushing, Equality* |

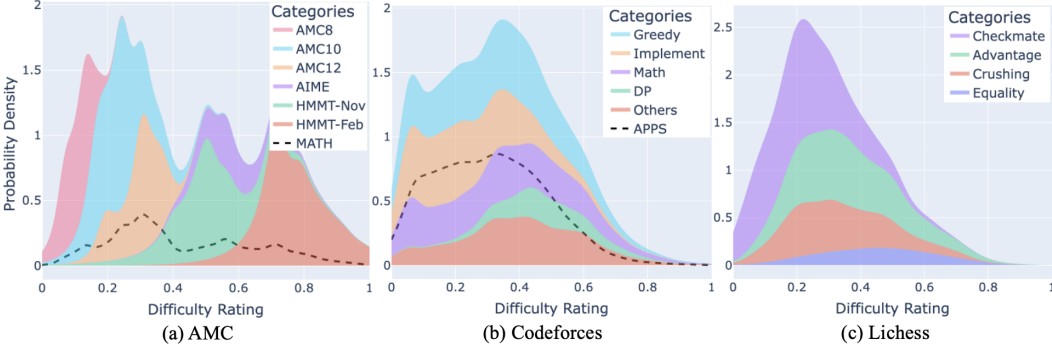

(a) AMC     (b) Codeforces     (c) Lichess

Figure 2: Distribution of difficulties in **E2H-AMC**, **E2H-Codeforces**, and **E2H-Lichess** datasets. Probability densities are colored by categories, showing their relative hardness. For **E2H-AMC** and **E2H-Codeforces**, we also draw the difficulty distribution of overlapping parts with the existing MATH [23] and APPS [22] datasets, respectively. The distributions reveal that our datasets include more challenging problems.

to test problem-solving skills in various domains. Three of these datasets — **E2H-AMC**, **E2H-Codeforces**, and **E2H-Lichess** — are newly curated, with difficulties estimated based on human performance statistics. The remaining datasets — **E2H-GSM8K**, **E2H-ARC**, and **E2H-Winogrande** — are widely used, and we provide additional sample-wise difficulty estimations derived from the performance of thousands of LLMs tracked on the Open LLM Leaderboard [5].

Accurate difficulty estimation is crucial for evaluating LLM performance across a spectrum of problems and for profiling its capabilities in easy-to-hard generalization. In the E2H-AMC dataset, item difficulty is represented by the percentage of students who answered each question correctly. For the E2H-GSM8K, E2H-ARC, and E2H-Winogrande datasets, the correctness of evaluator LLMs in answering the questions is recorded as binary values. In the E2H-Codeforces and E2H-Lichess datasets, the records indicate the success of users with specific capabilities in solving problems or items and include timestamps to reflect the variation in contestants' capabilities over time. However, deriving the item difficulty level from the raw statistics in these datasets is nontrivial. In this paper, we estimate the difficulty levels for each dataset sample using either IRT or Glicko-2 (with time consideration), an advanced version of the Elo rating system. Figure 1 shows example problems from the three newly curated datasets. The difficulty rating distributions of problems are plotted in Figure 2. The difficulty distributions of existing E2H-GSM8K, E2H-ARC, and E2H-Winogrande datasets are presented in Appendix I.

**Design Choices and Data Sources.** Easy2Hard-Bench integrates a variety of problems designed to assess abilities such as math, coding, puzzle solving, and reasoning — key areas where language models must excel beyond basic linguistic tasks. Our dataset spans various domains to provide a comprehensive assessment platform, addressing the gap in existing datasets that often neglect the progression from simple to complex problem-solving, which is critical for training adaptable and robust AI systems. We find data sources of problems with publicly available human performance statistics, which serve as a robust basis for difficulty estimation. We collect both high-quality problems at different levels of difficulty and rich real-world anonymous human performance data from popular online platforms including Art of Problem Solving, Codeforces), and Lichess.org.

**Data Sources for Difficulty Estimation.** The three platforms — Art of Problem-Solving, Codeforces, and Lichess — are popular and well-maintained forums in their respective domains. AMC/AIME/HMMT competitions publish the item difficulties per problem, i.e., the percentages of students who successfully solve each problem. Codeforces and Lichess provide user ratings using Elo/Glicko-2 algorithms. Meanwhile, *Open LLM Leadearboad* [5] open-sourced the detailed sample-wise evaluation results on over 5K LLMs. These reliable data sources (as summarized in table 1) ensure the reliability of our difficulty estimations, as they are continually viewed and scrutinized by the community. This validation by a broad audience confirms the high quality and reasonableness of the data used for estimating problem difficulty.

**Preprocessing Procedures.** After collecting problems, answers, and corresponding human or LLM performance statistics from the aforementioned dataset sources, we engage in a series of filtering and preprocessing steps to refine our dataset. **Filtering Procedures: (E2H-AMC)** We exclude problems whose answers cannot be succinctly represented as a unique LaTeX equation. This includes proof questions from HMMT or problems whose answers are lengthy expressions without a unique format. Additionally, problems that are inherently multiple-choice and lose context without their choices are also filtered out. **(E2H-Codeforces)** Our selection criteria focus on problems that feature high-quality Python3 solutions accompanied by extensive test cases. This helps mitigate known false positive issues of APPS [29]. **(E2H-Lichess)** We prioritize puzzles that can be answered with a single chess move, aligning with standard QA formats. Puzzles requiring multi-step solutions, which necessitate multi-turn QA and more complex evaluation metrics, are excluded. Furthermore, we ensure a uniform representation of puzzles across various categories and types through resampling. **Processing Steps: (E2H-AMC)** We convert all HTML and rich text elements into valid LaTeX strings, standardizing the LaTeX syntax, particularly within equation environments. **(E2H-Codeforces)** Solutions are formatted and code comments are removed. The validity of all test cases is checked. **(E2H-Lichess)** The provided FEN notations of puzzle chess boards are carefully converted into PGN and UCI notations, utilizing chess engines to generate move sequences from the start of the game to the puzzle situation. This conversion aims to provide LLMs with comprehensive information to solve the puzzles effectively. Evaluations from *Stockfish Chess Engine* [47] are also collected as additional information to provide to LLMs. More dataset preprocessing procedures are detailed in Appendix E.

## 2.1 Standardized Difficulty Estimation

**Method I: IRT without Time Consideration.** IRT [31] is utilized to estimate the difficulty of individual problems by analyzing the response patterns of participants. This model is particularly adept at handling datasets where the assumption of consistent participant ability over time is reasonable, such as in academic competitions like AMC.

Following prior work like [38], we applied different variations of IRT models including one-to-four parameter logistic models (denoted as 1PL-4PL models [38] and find that 1PL and 1PL-with-guessing (which is a simplified 3PL model; see Appendix A) works the best on our datasets.

The logistic model we used in IRT, specifically the 1PL-with-guessing model, is expressed as follows:

$$P(X_{ui} = 1 \mid \theta_u, b_i, c_i) = c_i + \frac{1 - c_i}{1 + e^{-(\theta_u - b_i)}},$$

where $P(X_{ui} = 1 \mid \theta_u, b_i, c_i)$ is the probability that user $u$ correctly solves problem $i$. $\theta_u$ represents the latent ability of user $u$. $b_i$ is the difficulty parameter of problem $i$. $c_i$ is the pseudo-guessing probability of problem $i$, which reflects that multiple-choice problems like those in E2H-AMC, E2H-ARC, and E2H-Winogrande may have a non-zero probability of solving just by randomly picking the choices.

We employ MCMC and variational Bayes ([34]) to fit the IRT models to the human item difficulty statistics in the **E2H-AMC** dataset and the sample-wise evaluation metrics of LLMs in **E2H-GSM8K**, **E2H-ARC**, and **E2H-Winogrande**. In a Bayesian framework, priors are typically assigned to the parameters $\theta_u$, $b_i$, and $c_i$ to facilitate the estimation process. Using Bayesian optimization, we also naturally obtain uncertainties on estimated difficulties. In general, IRT provides a robust measure of problem difficulty that reflects both the quality of the problem and the abilities of the participants.

**Method II: Glicko-2 System with Time Consideration.** The Glicko rating system [19] enhances the traditional Elo rating system [15] by incorporating a dynamic factor that accounts for the variability in player performance over time. This method is ideal for environments like Codeforces and Lichess,

where participants' abilities are not static and evolve based on their interaction with the problems over time, and is widely adopted in the player ranking of various sports [13, 53, 9].

Glicko-2 introduces the concept of rating deviation ($r_d$), measuring the reliability of a player's rating. Higher $r_d$ indicates greater uncertainty, which decreases as the player competes more frequently. Ratings are also adjusted for the time elapsed between contests, essential for accuracy in dynamic competitive environments. Moreover, Glicko-2 includes rating volatility $\sigma$, quantifying expected fluctuations in a player's rating. See Appendix A for mathematical details.

For the **E2H-Codeforces** and **E2H-Lichess** datasets that involve programming and chess puzzles, Glicko-2 can be used to treat each attempt on a problem as a discrete "game" where the problem itself can be thought of as one of the players. Therefore, by utilizing the human ratings provided by the data sources, we are able to calculate the problem difficulty ratings with deviations. This innovative application allows us to model problem difficulty as a dynamic entity that interacts with and adjusts to participants' changing abilities.

**Standardization of Estimated Difficulties.** Both IRT and Glicko-2 ensure a rigorous and comprehensive approach to difficulty estimation in Easy2Hard-Bench, providing a standardized complexity measure across varied domains. We conduct essential post-processing steps, such as IRT hyperparameter sweeping and model selection, outlier removal, and cross-reference with data sources' provided validation human ratings to ensure the reliability of our difficulty scores. The details are in Appendix E. We always normalize difficulty scores to $[0, 1]$ for standardization across datasets.

## 2.2 Verification of Difficulty Estimations

The quality and reliability of the Easy2Hard-Bench dataset hinge critically on the accuracy of our difficulty estimations compared to human perception. Verification of these estimations is thus essential. Further verification is deemed unnecessary for **E2H-AMC**, **E2H-Codeforces**, and **E2H-Lichess** datasets, where difficulty estimations are derived directly from well-established and highly publicized human performance metrics. The difficulty estimation of all these three datasets is based on the metrics of human participants in real evaluation and the ratings of human participants by well-accepted rating systems acknowledged by large professional communities (for example, in Appendix F we compare the estimated difficulty with professional guides, e.g., contest difficulty rating on AoPS and justify the natural well-alignment; see Appendix F for E2H-Codeforces and E2H-Lichess). However, further verification is imperative for **E2H-GSM8K, E2H-ARC, and E2H-Winogrande** datasets, which are primarily based on model performance statistics, to ensure that these metrics accurately reflect human performance potential.

**Human Verification of E2H-GSM8K, E2H-ARC, E2H-Winogrande Difficulties.** To validate our model-based difficulty estimations, we conduct surveys where participants are asked to rank problem difficulties. We show participants pairs of problems and ask them to determine which of the two is more difficult. In our survey, we show the participants 10 pairs of problems from each of the 3 datasets and request they rank the difficulty of two questions in each pair. The majority vote from these surveys is then compared to our model-derived difficulty rankings to assess alignment. For each pair of problems, we use the majority vote from 5 participants' responses as the human's opinion and compare it with the rank based on our relative estimated difficulty.

We outline the verification procedure below:
- *Step 1: Compute Per-Pair Discrepancy.* We define the discrepancy for the $i$-th pair of problems as:

$$\delta^{(i)} = |s_h^{(i)} - s_e^{(i)}| \times \mathbb{1}\{s_h^{(i)} < s_e^{(i)}\}$$

where $\delta^{(i)}$ is non-zero only if the computed difficulty $s_h^{(i)}$ for what is perceived by experts as the harder problem is actually lower than $s_e^{(i)}$, the easier one. This metric captures instances and magnitudes of disagreement between our dataset and expert judgment.
- *Step 2: Report Discrepancies.* We provide statistical and visual summaries of the discrepancies. The mean and standard deviation of $\{\delta^{(i)}, \forall i \in [n]\}$ offer a sense of the average discrepancy and its variability. A histogram of $\{\delta^{(i)}, \forall i \in [n]\}$ visually represents the distribution of these discrepancies, highlighting the frequency of significant misalignment.

The per-pair rank matching accuracies $\mathbb{1}\{s_h^{(i)} < s_e^{(i)}\}$ human evaluation results and our estimation based on IRT are reported in Table 3. This verifies the well-alignment between the difficulty ratings

Table 3: Verification of estimated difficulties on E2H-GSM8K, E2H-ARC, and E2H-Winogrande, which are based on collective statistics of LLMs and obtained using Item Response Theory (IRT). IRT-estimated difficulties align well with human preferences and outperform the alignment with GPT4.

| | Metric | E2H-GSM8K | E2H-ARC | E2H-Winogrande |
|---|---|---|---|---|
| IRT v.s. Human | Matching Acc. | $0.942 \pm 0.046$ | $0.737 \pm 0.047$ | $0.734 \pm 0.040$ |
| | Avg. Per-pair Discrepancy | $0.026 \pm 0.015$ | $0.096 \pm 0.020$ | $0.113 \pm 0.027$ |
| IRT v.s. GPT4 | Matching Acc. | $0.836 \pm 0.035$ | $0.616 \pm 0.028$ | $0.597 \pm 0.032$ |
| | Avg. Per-pair Discrepancy | $0.057 \pm 0.004$ | $0.137 \pm 0.007$ | $0.204 \pm 0.008$ |

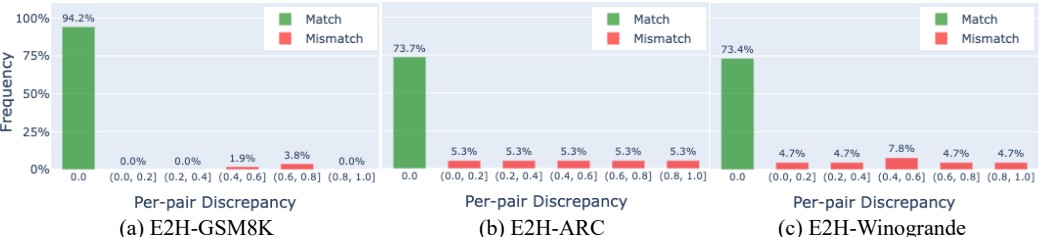

(a) E2H-GSM8K     (b) E2H-ARC     (c) E2H-Winogrande

Figure 3: Histograms show the frequency of sample pairs ordered by human judges across **E2H-GSM8K**, **E2H-ARC**, and **E2H-Winogrande** datasets. Green bars indicate pairs where our difficulty labels match human judgment; red bars indicate mismatches. Most labels align with human preferences. Inconsistent pairs are uniformly distributed across different discrepancy ranges. Examples are listed in Appendix F.

estimated by IRT using a large set of LLMs performance statistics (on *Open LLM Leaderboard* [5]) and the human consensus. Reflecting the possibilities of using a large collection of LLMs to serve as a surrogate of human crowds on certain tasks to study collective behavioral statistics.

**Discrepancy between LLM-Sourced Ratings and Human Verification.** It is crucial to understand the discrepancy between LLM-sourced difficulty ratings and human-verified ratings on the E2H-GSM8K, E2H-ARC, and E2H-Winogrande datasets. We provide detailed analyses of where model estimations align or diverge from human evaluations. Differences are visually represented in histograms, highlighting the extent and nature of these discrepancies. With high matching accuracy and low average $\delta^{(i)}$, the IRT method behaves better in alignment on E2H-GSM8K. We attribute the gap among datasets to the intrinsic difference of domain. The amount of reasoning during solving arithmetic problems in E2H-GSM8K is more measurable than ARC based on natural science knowledge and Winogrande focusing on one-step commensense reasoining. Human subjects are more sensitive to the difficulty in E2H-GSM8K. During our human evaluation, many participants reported that they could not easily rank difficulties on pairs of E2H-Winogrande and E2H-ARC problems, but this did not happen on E2H-GSM8K. In Appendix F, we provide problem examples and corresponding human-, model-performance statistics and IRT results.

**Scaling-up via LLM-as-a-Judge.** While human evaluation is the gold standard, our limited pool of only 50 participants constrains the verification scale. To expand verification, we also employ LLMs as proxies to increase the number of tests [56]. We ranked 2,000 pairs of problems with *GPT4-Turbo* and careful prompting (Appendix F) and 3 majority votes per pair. The comparison between GPT4's ranking and IRT results is also reported in Table 3. Interestingly, we find our IRT results align better with humans compared to GPT4. And the GPT4 rankings are not well-aligned with humans on E2H-ARC and E2H-Winogrande; see Appendix F for discussions.

## 3 Benchmarking SoTA LLMs via Easy2Hard-Bench

**Model Selections and Details.** In light of the novel and challenging problems presented in our Easy2Hard-Bench, we have selected the most advanced generations and versions from both proprietary and open-source large language model (LLM) families for evaluation. Our lineup includes *GPT4-Turbo* [1], *Claude3-Opus* [2], and *Gemini1.5-Pro* [36] from the proprietary series, alongside

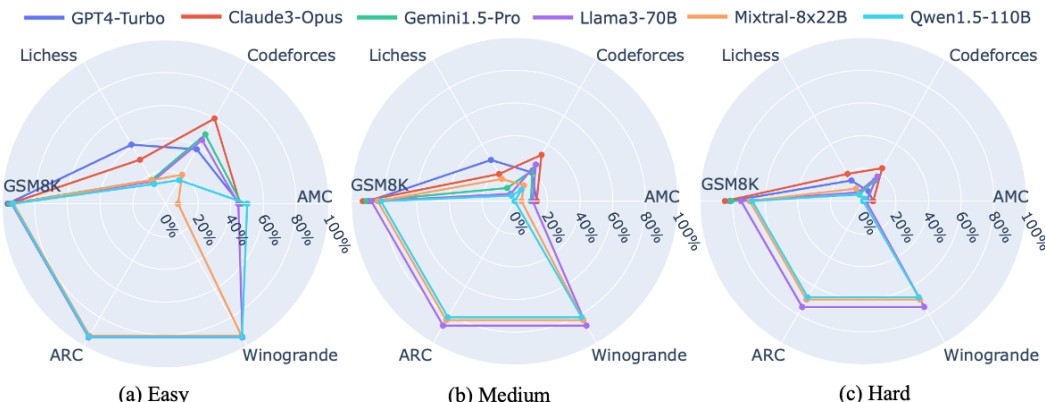

Figure 4: Radar plots show the performance of six state-of-the-art LLMs on the six Easy2hard-Bench datasets, divided into easy, medium, and hard parts by difficulty rating (equal quantiles). The newly curated datasets (**E2H-AMC**, **E2H-Codeforces**, **E2H-Lichess**) are more challenging than the others. Overall performance of all models significantly decreases with increasing difficulty.

*Llama3-70B* [32], *Mixtral-8x22B* [27], and *Qwen1.5-110B* [3] from the open-source series. By leveraging these state-of-the-art LLMs, we aim to gain a deeper understanding of AI capabilities to solve problems of increasing difficulty across different domains. The selection of these models ensures that even on the most challenging problems, their performance provides valuable insights, allowing us to assess the capabilities of these LLMs across a spectrum of difficulties comprehensively.

**Evaluation Setups and Metrics.** Our evaluation setups predominantly adopt metrics from existing evaluation pipelines, as the evaluation design for assessing math, coding, and reasoning tasks is thoroughly established. An exception is made for chess puzzles, a relatively unexplored challenge for LLMs, necessitating a specifically tailored evaluation setup and prompt template (see appendix G). While techniques like chain-of-thought prompting [50] and majority voting [49] can enhance LLM performance, our focus remains on benchmarking datasets and difficulty ratings with naive zero- or few-shot prompting setups. We defer some more complex setups to Appendix G and future work.

- **E2H-AMC:** Similar to MATH [23], problems and answers are encoded as LaTeX strings, with the final answer required to be enclosed in "□". Accuracy of matching the answer within "□"s is reported.
- **E2H-Codeforces:** Following the evaluation frameworks of HumanEval [8] and APPS [22], our evaluation package supports metrics like "pass@k". However, due to resource constraints, in the paper we mainly focus on the test case average accuracy, a standard from APPS.
- **E2H-Lichess:** Chess puzzles are converted into QA format. Prompts are designed to describe the puzzle using multiple chess notations including FEN, PGN, and UCI. Moreover, evaluations and annotations from the *Stockfish Chess Engine* [47] are provided. LLMs are asked to format their answers, the next chess moves, in both PGN or UCI notation. The answer matching criterion detailed in Appendix G ensure that correct answers in either notation can be recognized.
- **E2H-GSM8K, E2H-ARC, and E2H-Winogrande:** For these existing datasets, we adhere to the standard evaluation protocols used by *Open LLM Leaderboard* [5], which applies 5-, 25-, and 5-shot prompting for E2H-GSM8K, E2H-ARC, and E2H-Winogrande, respectively. As E2H-ARC and E2H-Winogrande are multiple-choice QAs requiring log-probabilities from LLMs, proprietary models cannot be evaluated on them.

**Profiling of Performances over Ranges of Difficulties.** We begin by presenting the performance of LLMs on all Easy2Hard-Bench datasets, segmented into easy, medium, and hard difficulty levels. Results are illustrated in Figure 4 through radar plots for each model across the six datasets, effectively visualizing performance on different problem domains. It is evident that performance notably decreases as difficulty increases, validating the effectiveness of our difficulty estimations. The radar plots (Figure 4) further show that the newly curated datasets (**E2H-AMC**, **E2H-Codeforces**, **E2H-Lichess**) are much more challenging than the pre-existing ones, because they extend the difficulty range greatly compared to existing selections.

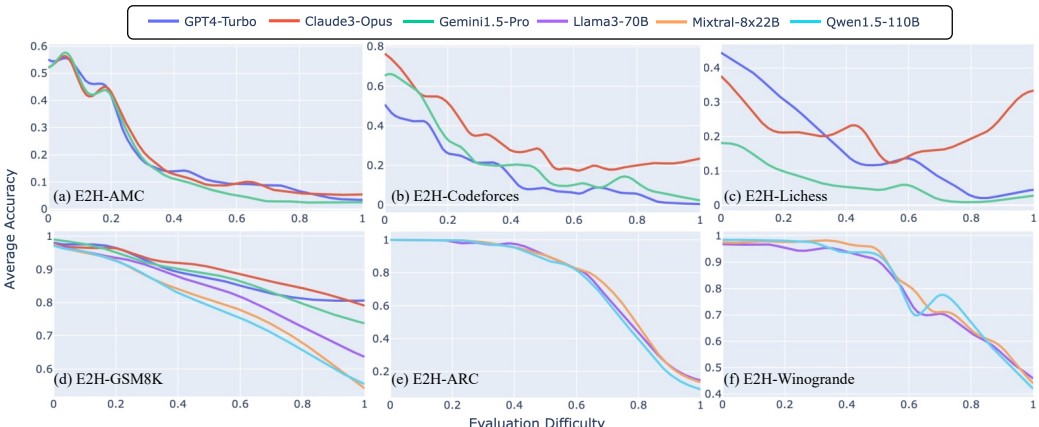

Figure 5: Model performance on varying difficulties. Six line-plots show the performance of six SoTA models across six datasets in Easy2hard-Bench. As evaluation difficulty increases, most models show monotonic decreasing accuracies, validating the correctness of provided difficulty ratings. Different model performances may diverge at higher difficulties, as observed in **E2H-GSM8K**, or remain closely matched, as seen in **E2H-ARC** and **E2H-Winogrande**. For **E2H-AMC**, **E2H-Codeforces**, and **E2H-Lichess datasets**, 0-shot inference was utilized. Notably, *GPT4-Turbo*, *Claude3-Opus*, and *Gemini1.5-Pro* exhibit similar mathematical capabilities, though their performance in chess and 0-shot code ability varies. Specifically, *Claude3-Opus* performs well on relatively hard chess puzzles, suggesting its training data may have had greater exposure to chess content.

Detailed performance trends are then analyzed through line plots that show model behavior against increasing difficulty levels for each dataset (Figure 5). Thanks to the continuous difficulty rating and accompanying uncertainty for each problem, we can plot smooth average performance curves. This granularity allows us to observe that while performance generally declines with difficulty, the extent of this decline varies significantly among models and datasets. In particular, performance disparities between models may become more pronounced with higher difficulty levels, especially noticeable in datasets like **E2H-GSM8K**. Conversely, in datasets such as **E2H-ARC** and **E2H-Winogrande**, performance differences are small across difficulties despite overall fluctuations. On the E2H-Lichess dataset, while *GPT4-Turbo* excels at simpler puzzles, *Claude3-Opus* demonstrates superior performance on the more complex puzzles, even reversing the trend of declining performance. On GSM8K, *Claude3-Opus* shows the most gradual decline in performance, whereas *Qwen1.5-110B* exhibits the steepest drop.

## 4  Profiling Easy2Hard Generalizations

Contrary to other LLM benchmarking suites like [44], the Easy2Hard-Bench provides sample-wise continuous difficulty ratings with uncertainty for all six datasets, enabling a big step forward in benchmarking LLM capabilities and profiling their behaviors. Instead of only assessing the static behavior of specific checkpoints, our approach allows for fine-grained profiling of LLMs as they generalize across various training and evaluation difficulties. This also caters to the need to simulate challenging problems like weak-to-strong generalization [7]. To our best knowledge, Easy2Hard-Bench is the first to deliver detailed easy-to-hard generalization results across continuous, wide-range of difficulties on LLMs.

**Method to Profile Easy2Hard Generalizations over Ranges of Training and Evaluation Difficulties.** To capture the "two-dimensional" generalization behavior, we divide the training data into $a$ bins based on difficulty ratings and undertake training $a + b$ times: $a$ times on each difficulty bin and $b$ times on randomly chosen subsets of the same size. During evaluation, we assess all $a + b$ trained LLMs across the complete range of evaluation difficulties. We further interpolate the evaluation performances of the $a$ LLMs trained at different difficulty levels, by employing an RBF kernel. We also subtract the "background performance" of the $b$ LLMs trained on random difficulties

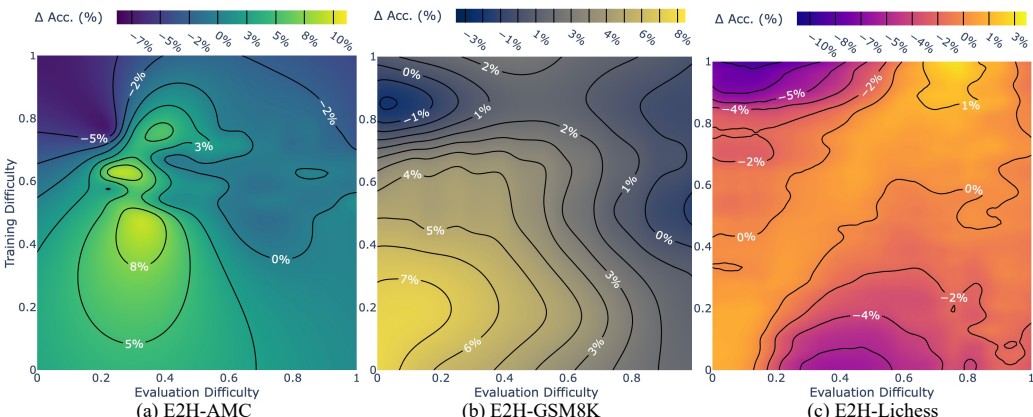

Figure 6: Easy2hard generalization performance across varying training and evaluation difficulties. Heat maps illustrate the easy-to-hard generalization performance on three datasets: **E2H-AMC**, **E2H-GSM8K**, and **E2H-Lichess**, using different LLMs and setups. LLMs were trained on subsets of training splits of varying difficulty (y-axis) via Supervised Fine-Tuning (SFT) and were evaluated across all evaluation difficulties (x-axis). The color gradient represents the performance difference relative to models trained on randomly selected difficulties of same sizes. We observe **(1)** generalization benefits when training and evaluation difficulties are similar, and **(2)** training on more challenging samples poses increased generalization difficulties.

and thus highlight the generalization gain. The results are visually represented through contour plots in Figure 6.

**Experimental Setups.** In our preliminary experimental exploration, we focus on Supervised Finetuning (SFT) with relatively smaller LLMs, while deferring more specialized finetuning frameworks for future studies. We deploy three setups on the **E2H-AMC**, **E2H-GSM8K**, and **E2H-Lichess** datasets, setting $a = 7$ and $b = 1$ unless specified otherwise.

- **E2H-AMC and E2H-GSM8K:** We utilize *GPT3.5-Turbo* [1], still a leading proprietary LLM with accessible fine-tuning APIs. On E2H-AMC, where the training split is relatively small (1,000 training and 2,975 evaluation samples), we invert the roles of train and eval splits.
- **E2H-Lichess:** We employ a novel approach by utilizing *GPT2* models, which have been retrained on a vast corpus of real-world chess games sourced from E2H-Lichess (not overlapping with the puzzles). To better capture the nuances of chess move notations, we replace the standard tokenizer with a specialized one designed specifically for chess moves, similar to the strategy employed in [39]. The optimal pretrained *GPT2* checkpoint is sourced from LeonLLM [30], and used to profile the easy-to-hard generalization on the Lichess puzzles in Easy2Hard-Bench.

**Observations on Easy2Hard Generalization Behaviors.** In Figure 6, we present the easy-to-hard generalization margins through contour plots. Across different datasets and models, a common pattern emerges in the generalization behavior: a "generalization ridge" typically aligns with the diagonal, where training and evaluation difficulties are similar. For E2H-GSM8K, this generalization gain diminishes as the training difficulty increases, aligning with findings reported in [20]. E2H-AMC and E2H-GSM8K exhibit much poorer generalization when trained on more difficult samples compared to baselines trained on mixed difficulties. However, with the smaller *GPT2* models and a tailored tokenizer for E2H-Lichess, the generalization ridge extends further across difficulties.

We hypothesize that for large LLMs like *GPT3.5-Turbo*, especially with complex real-world problems like those in E2H-AMC, it is generally challenging to generalize effectively using frameworks such as SFT, and that naive easy-to-hard generalization behavior deteriorates as difficulty escalates. This observation of poor scaling law aligns with findings reported in [7]. We conclude the exploration with an open question regarding the development of better LLM training paradigms that could improve the scalability of training on increasingly difficult problems, aiming for advancements towards close-to-human and super-human LLM performance in the near future.

## Acknowledgements

Ding, Deng, Agrawal and Huang are supported by DARPA Transfer from Imprecise and Abstract Models to Autonomous Technologies (TIAMAT) 80321, National Science Foundation NSF-IIS-2147276 FAI, DOD-ONR-Office of Naval Research under award number N00014-22-1-2335, DOD-AFOSR-Air Force Office of Scientific Research under award number FA9550-23-1-0048, DOD-DARPA-Defense Advanced Research Projects Agency Guaranteeing AI Robustness against Deception (GARD) HR00112020007, Adobe, Capital One and JP Morgan faculty fellowships. A. Anandkumar is supported by the Bren chair professorship and AI 2050 Schmidt Sciences senior fellowship.

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

# Appendix

## A Difficulty Estimation Details

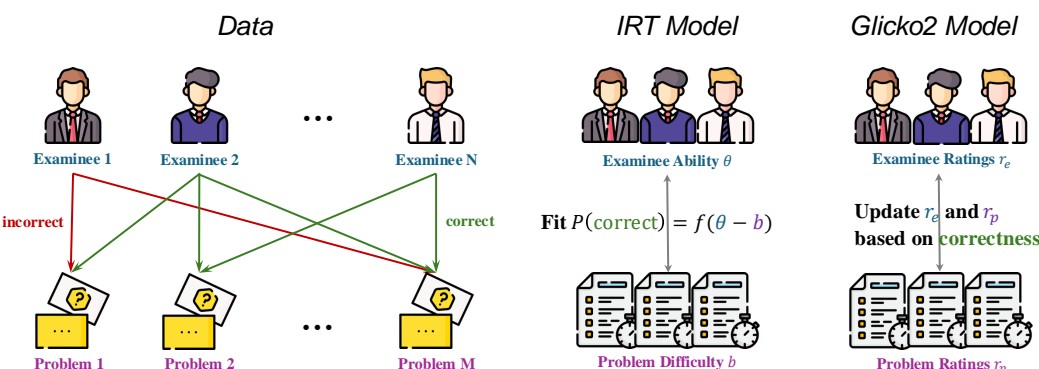

Figure 7: Overview of difficulty estimation framework utilizing performance data and statistical models. *Left*: Input data matrix showing binary performance records (correct/incorrect) for $N$ examinees across $M$ problems. *Right*: Two parallel modeling approaches - Item Response Theory (IRT) estimates examinee ability ($\theta$) and problem difficulty ($b$) parameters through logistic fitting of $P(\text{correct})$, while the Glicko2 rating system dynamically updates both examinee ($r_e$) and problem ($r_p$) ratings based on performance outcomes. These complementary methods enable robust difficulty quantification from empirical solve attempts.

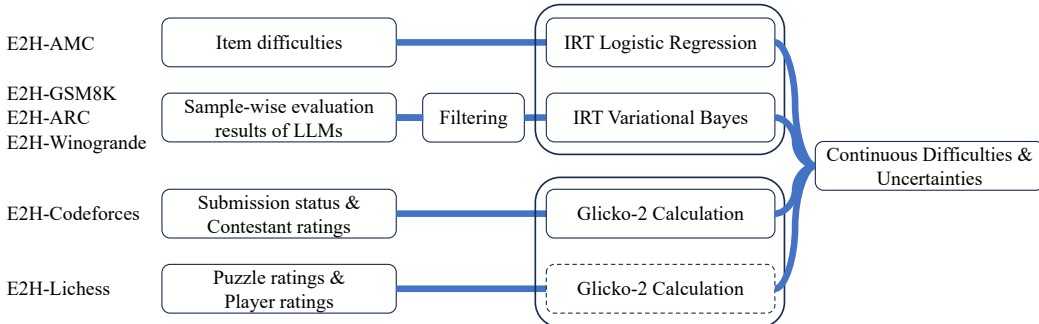

Figure 8: Comprehensive workflow depicting the difficulty estimation process for Easy2Hard-Bench datasets. The pipeline encompasses four types of input sources: existing item difficulties, sample-wise LLM evaluation results, competition submission data, and game ratings, which are processed through IRT and Glicko-2 statistical models to derive continuous difficulty scores with associated uncertainties.

### A.1 Preprocessing to IRT/Glicko-2 Inputs

**E2H-AMC.** We collect the item difficulty directly or indirectly from the official reports. Item difficulty refers to the percentage of participants answering an item correctly. MAA provides item difficulties of AMC and AIME directly. HMMT presents the score of each individual or team on each problem, with which we can compute the corresponding item difficulty.

**E2H-Codeforces.** We collect the submission records and contestant rating history via the official API. The submission record shows that whether the specific submission is accepted or not. The rating history illustrates the variation of a contestant's performance. Moreover, we scrape the official rating for problems, and we use it as an alignment of our estimation.

**E2H-Lichess.** We gather the puzzle rating and the player ratings in each puzzle. Puzzle rating shows the difficulty of the puzzle approximately while player ratings indicates the fluctuation of strength.

**E2H-GSM8K, E2H-ARC, E2H-Winogrande.** We gather the evaluation results of LLMs on each problem from these datasets reported in open LLM leaderboard. For each dataset, we use a greedy search algorithm to find a subset of LLMs so that the difficulty ranking results based on the average accuracy of these models is as near as possible to human verification results.

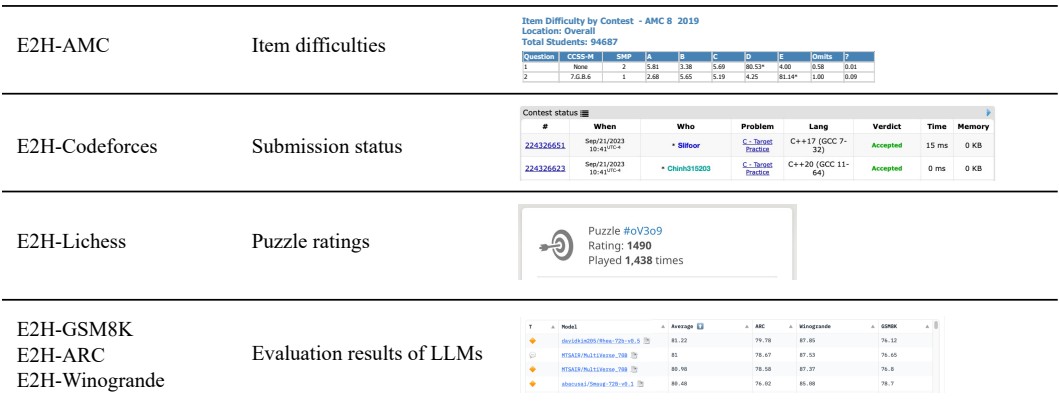

| E2H-AMC | Item difficulties | |
| E2H-Codeforces | Submission status | |
| E2H-Lichess | Puzzle ratings | |
| E2H-GSM8K E2H-ARC E2H-Winogrande | Evaluation results of LLMs | |

Figure 9: Representative examples of problem inputs collected from source platforms for difficulty estimation. Each screenshot demonstrates the original presentation format and user interface elements as they appear to examinees.

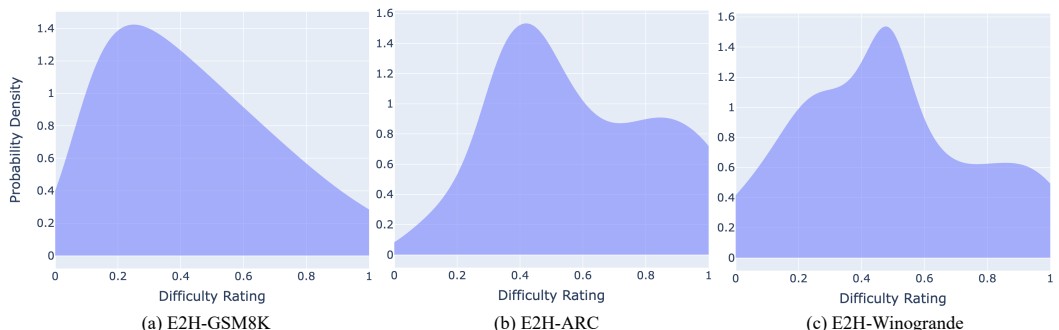

(a) E2H-GSM8K  (b) E2H-ARC  (c) E2H-Winogrande

Figure 10: Distribution of difficulties in E2H-GSM8K, E2H-ARC, and E2H-Winogrande. Probability densities are colored by categories, showing their relative hardness.

## A.2 IRT

IRT [31] is utilized to estimate the difficulty of individual problems by analyzing the response patterns of participants. It is based on the idea that the probability of a correct response to an item (problem) is a logistic function of some person and item parameters. Since item difficulty is one of item parameters in IRT method, we aim to estimate the difficulty of problems by fitting the IRT model to the performance metrics we collect previously.

The logistic model we used in IRT, specifically the 1PL-with-guessing model, is expressed as follows:

$$P(X_{ui} = 1 \mid \theta_u, b_i, c_i) = c_i + \frac{1 - c_i}{1 + e^{-(\theta_u - b_i)}},$$

where $P(X_{ui} = 1 \mid \theta_u, b_i, c_i)$ is the probability that user $u$ correctly solves problem $i$. $\theta_u$ represents the latent ability of user $u$. $b_i$ is the difficulty parameter of problem $i$.

To illustrate how we choose the parameters of IRT model for difficulty estimation, we compare four variations of IRT models (denoted as 1PL-4PL models) with different number of logistic model parameters.

- **1PL:** The model assumes that guessing is included in the ability and all items fitting the model sharing the same discrimination. So the only parameter to describe the items is $b_i$, i.e.,

$$P(X_{ui} = 1 \mid \theta_u, b_i) = \frac{1}{1 + e^{-(\theta_u - b_i)}}.$$

- **2PL**: The model assumes that guessing is included in the ability but the item $i$ fitting the model has the discrimination $a_i$. So the parameters to describe the items are $a_i$ and $b_i$, i.e.,

$$P(X_{ui} = 1 \mid \theta_u, a_i, b_i) = \frac{1}{1 + e^{-a_i(\theta_u - b_i)}}.$$

- **3PL**: The model assumes that guessing is excluded in the ability and formulated as an asymptotic minimum of $c_i$ for each item. So the parameters to describe the items are $a_i$, $b_i$ and $c_i$, i.e.,

$$P(X_{ui} = 1 \mid \theta_u, a_i, b_i, c_i) = c_i + \frac{1 - c_i}{1 + e^{-a_i(\theta_u - b_i)}}.$$

- **4PL**: Besides the guessing formulated as $c_i$, the model assumes that the item is intrinsically unsolvable with probability and the corresponding asymptotic maximum is formulated as $d_i$. So the parameters to describe the items are $a_i$, $b_i$, $c_i$ and $d_i$, i.e.,

$$P(X_{ui} = 1 \mid \theta_u, a_i, b_i, c_i, d_i) = c_i + \frac{d_i - c_i}{1 + e^{-a_i(\theta_u - b_i)}}.$$

For our case, the problems from both math competitions (E2H-AMC) and prevalent reasoning task dataset (E2H-GSM8K, E2H-ARC, E2H-Winogrande) are well-defined and principally solvable, suggesting that $d_i = 1$, and there is no discrimination among different problems within the same dataset, suggesting that $b_i = 1$. Therefore, we use only two parameters, difficulty $b_i$ and guessing $c_i$, in our difficulty estimation and propose 1PL-with-guessing (**1gPL**) as following

$$P(X_{ui} = 1 \mid \theta_u, b_i, c_i) = c_i + \frac{1 - c_i}{1 + e^{-(\theta_u - b_i)}}.$$

For E2H-AMC, the input examples for difficulty estimation are exactly item difficulties of problems released in the official reports, thus we do not need any further preprocessing. Moreover, IRT implicitly assumes that users consistently solve problems. Although a student usually participants in the contest at the specific level only once, we assume that the ability of students taking contests at the same level in different years is constant.

## A.3 Glicko-2

The update mechanism in the Glicko-2 system incorporates the outcome of games, the reliability of the rating, and the time between games as follows:

$$r' = r + \frac{q}{\frac{1}{r_d^2} + \frac{1}{d^2}} \sum_{j=1}^{n} g((r_d)_j)(s_j - \mathbb{E}(s_j|r, r_j)), \quad r'_d = \sqrt{\frac{1}{\frac{1}{r_d^2} + \frac{1}{d^2}}}$$

where $r'$ and $r'_d$ are the updated rating and rating deviation, respectively. $q = \log 10/400$ is a scaling factor. $g(RD) = 1/\sqrt{1 + 3q^2(RD^2)/\pi^2}$ is a function that reduces the impact of matches with opponents having high rating deviations. $s_j$ represents the outcome of game j (1 for a win, 0.5 for a draw, 0 for a loss). $\mathbb{E}(s_j|r, r_j)$ is the expected score against opponent $j$, who has rating $r_j$ and deviation $RD_j$. $d^2$ is the variance of the rating changes, $d^2 = 1/(q^2 \sum_{j=1}^{n} g(RD_j)^2 \mathbb{E}(s_j|r, r_j)(1 - \mathbb{E}(s_j|r, r_j)))$.

- **Step 1: Ancillary quantities.** During each rating period (such as the interval between contests), consider a player with current rating $\mu$ and rating deviation $\phi$. Assuming that this player plays against $m$ opponents with ratings $\mu_1, \cdots, \mu_m$ and rating deviations $\phi_1, \cdots, \phi_m$ and these games result in scores $s_1, \cdots, s_m$, we compute two ancillary quantities $v$ and $\Delta$:

$$v = \left[ \sum_{j=1}^{m} g(\phi_j)^2 \mathbb{E}[s|\mu, \mu_j, \phi_j]\{1 - \mathbb{E}[s|\mu, \mu_j, \phi_j]\} \right]^{-1}, \quad \Delta = v \sum_{j=1}^{m} g(\phi_j)\{s_j - \mathbb{E}[s|\mu, \mu_j, \phi_j]\}$$

where

$$g(\phi_j) = \frac{1}{\sqrt{1 + 3\phi_j^2/\pi^2}}, \quad \mathbb{E}[s|\mu, \mu_j, \phi_j] = \frac{1}{1 + \exp\{-g(\phi_j)(\mu - \mu_j)\}}.$$

- **Step 2: Rating volatility.** The second step is to update rating volatility $\sigma$. This parameter measures the expected fluctuation of rating over time. A larger $\sigma$ means that the player behaves more inconsistently across the past rating periods. With a small constant $\tau$ constraining the volatility over time, we use the iterative procedure to find the solution $x_0$ for $f(x) = 0$ where $f$ is given by

$$f(x) = \frac{e^x(\Delta^2 - \phi^2 - v - e^x)}{2(\phi^2 + v + e^x)^2} - \frac{x - 2\ln\sigma}{\tau^2},$$

  and set the new rating volatility as $\sigma' = \exp(x_0/2)$.
- **Step 3: Rating and rating deviation.** With the new rating volatility $\sigma'$, we calculate the updated rating deviation $\phi'$ and the updated $\mu'$ as follows

$$\phi' = \frac{1}{\sqrt{\frac{1}{\phi^2 + \sigma'^2} + \frac{1}{v}}}, \ \mu' = \mu + \phi'^2 \sum_{j=1}^{m} g(\phi_j)\{s_j - \mathbb{E}[s|\mu, \mu_j, \phi_j]\}.$$

For the problems in $E2H - Codeforces$, For the problems in $E2H - Lichess$, since Lichess uses the Glicko-2 system to rate the players and the puzzles, so we inherit the puzzle ratings for the original problems and convert them into $[0, 1]$ scale.

## B  Related Works

Our work intersects with several established datasets and benchmarks across different domains, each challenging and assessing specific areas of capabilities of large language models. First, we provide an overview of some notable prior works on related datasets and benchmarks, categorized by domain.

**Math LLM Benchmarks:** MATH [23] offers a variety of high-school level math problems, with a small proportion overlapping with our E2H-AMC dataset in the Easy2Hard-Bench, since MATH also collects problems from math competitions like AMC8 and AMC10 (see fig. 2). However, MATH also includes a large portion of easier math problems. MATH offers a coarse five-level difficulty rating for each problem. Agieval [57], SciBench [48], MiniF2F [55], and OlympiadBench [21] generally aim towards challenging, math competition-style problems. However, to address the lack of a sufficient number of problems in math competitions, they mix in other sources like the US's SAT and Chinese GaoKao questions, or problems from other science subjects like physics and chemistry into the dataset. As a result, these datasets do not maintain a continuous and uniform span of difficulty nor a clear and recognizable concept of difficulty. On the other hand, apart from focusing on math problem solving, GHOSTS [17] proposes a mixture of five types of abstract mathematical challenges. However, it only has 709 questions and requires professional expert evaluation. Recently, [51] proposes LeanDojo, a toolkit and playground for LLM theorem proving.

**Coding LLM Benchmarks:** APPS [22] is one of the earliest benchmarks for evaluating machine learning models on code generation, featuring 10,000 problems with performance assessed against multiple test cases. HumanEval [8] is a dataset for code synthesis from docstrings, revealing insights and achieving notable problem-solving rates through repeated sampling strategies, popularizing the pass@k metric. LiveCodeBench [25] proposes continuously incorporating new problems from coding competitions, aiming to provide a more holistic assessment. Only a fraction of problems in LiveCodeBench have difficulty ratings provided by specific code platforms; however, some difficulty ratings are categorical, and ratings from different sources are not properly unified and standardized. TACO [28] introduces more fine-grained problem tagging, yet only part of the problems has a coarse five-level difficulty rating.

**Common-Sense Reasoning LLM Benchmarks:** HellaSwag [54] employs adversarial filtering to challenge models with commonsense inference, where machines lag significantly behind humans. OpenBookQA [33] tests multi-hop reasoning on elementary science facts, uncovering significant gaps between human and AI capabilities. WinoGrande [40] enhances the Winograd Schema with a larger, bias-reduced dataset, critically evaluating commonsense reasoning in AI. ARC [11] challenges AI with complex science questions beyond current model capacities. BoolQ [10] and PIQA [6] expose the limitations of pretrained models in answering natural yes/no questions and physical commonsense queries, respectively, highlighting the discrepancies in real-world reasoning abilities. None of these datasets have fine-grained or continuous difficulty ratings.

To the best of our knowledge, there are very few publicly available established LLM datasets and benchmarks on puzzles (e.g., chess, go, maze, sudoku, etc.). Therefore, instead of reviewing datasets, we focus on reviewing the methodological works on LLMs for puzzles, which may or may not have publicized the dataset used for training.

**LLMs for Puzzles:** Move-by-move [26] introduces a novel large-scale dataset comprising over 298K chess move-commentary pairs from 11K games, focusing on generating natural language descriptions that capture diverse commentary styles and the pragmatic context of each move. Meanwhile, Chess Transformer [35] leverages a massive training corpus of 2.8 million games in Portable Game Notation, fine-tuning OpenAI's GPT-2 to generate strategic chess moves and recognize classic game formations, demonstrating the model's capacity to engage in strategic thinking beyond mere move generation. In a more integrated approach, ChessGPT [16] bridges policy learning and language modeling by incorporating a large-scale game and language dataset, enhancing decision-making in chess with combined insights from historical games and analytical strategies. Google Brain's Grandmaster [39] significantly scales up, training a 270M parameter transformer on a dataset annotated with 15 billion data points from 10 million games, evaluated by the Stockfish [47] engine, achieving high-level performance that challenges conventional chess engines and even surpasses AlphaZero's networks in certain aspects without domain-specific adaptations.

**LLM Benchmarks with Difficulty Annotations:** Besides the previously mentioned datasets, several other LLM benchmarks annotate each problem with difficulty, though these annotations are typically categorical or presented pairwise. GPQA [37], a dataset consisting of graduate-level multiple-choice questions, utilizes the average of a 4-point difficulty rating provided by two expert validators. The medical concepts question answering benchmark, MedConceptsQA [43], assigns difficulty levels to questions based on the distances among four options in the medical code vocabulary hierarchy, represented as an undirected graph. In this setup, options in harder questions are closely related due to smaller distances. The linguistic reasoning benchmark LingOly [4] categorizes question difficulty into five levels based on semantic similarity to English and reasoning complexity. OlympicArena [24], a comprehensive cognitive reasoning benchmark, categorizes difficulty into three levels, evaluated by LLMs based on the required abilities for problem-solving, ranging from direct recall of facts to logical or visual reasoning. All difficulty annotations in these benchmarks are categorical. Meanwhile, ConsisEval [52] comprises pairs of questions ordered strictly by difficulty; in each pair, the easy problem is sourced from existing datasets, while the hard problem is derived from the easy one through either human annotation or automatic generation.

As an LLM benchmarking and evaluation suite, we share similarities with other LLM evaluation suites in aspects such as the design of evaluation pipelines and methods. We review some prior work on LLM evaluation methods as follows.

**LLM Evaluation Suites and Methods:** The Open LLM Leaderboard [5] utilizes EleutherAI's Evaluation Tool to benchmark LLMs across diverse tasks, emphasizing realistic performance assessments. AlpacaEval 2.0 [14] introduces regression analysis to mitigate biases in LLM auto-evaluations, enhancing alignment with human judgments. MT-Bench [56] employs LLMs as judges for multi-turn evaluations on crowdsourced platforms, effectively approximating human preferences. Nevertheless, they generally lack a domain-specific approach with progressively scaled difficulty, which is critical for detailed assessments of LLMs' learning curves and adaptability.

Finally, our work also explores the generalization behaviors of LLMs, especially under the easy-to-hard setup. We review the prior work on easy-to-hard generalization below.

**Easy2Hard Generalization:** [42] explores how recurrent neural networks generalize from simple to complex tasks by increasing computational steps. [41] introduces datasets spanning various difficulties to study this generalization capability in tasks from prefix sums to chess puzzles, which are also sourced from Lichess [46]. On LLMs, [20] studies show that pretrained language models can generalize well from easy to hard data using simple fine-tuning methods, often matching or exceeding the performance of models fine-tuned on hard data. This suggests that collecting easy data may be more beneficial for fine-tuning than attempting to label noisier, costlier hard data. However, [20] experimental study is limited by the coarse difficulty levels and heuristically chosen hardness measures.

## C  Limitations

This section summarizes the major limitations of this work. This work primarily focuses on dataset and benchmark creation, which brings about several limitations.

- In our evaluations, we only consider primary setups and metrics because we are mainly a dataset and benchmark work. We did not explore many state-of-the-art setups or methods, such as chain of thought for evaluation. This focus on fundamental approaches might limit the usefulness and comprehensiveness of our findings regarding the latest evaluation techniques.
- We conducted a human evaluation to verify the estimated difficulty on three datasets in Easy2Hard-Bench: E2H-GSM8K, E2H-ARC, and E2H-Winogrande. We involved 50 participants to rank 100 pairs of problems per dataset. Due to time and resource constraints, we could only secure a limited number of participants for the human evaluation. This limited scale affects the robustness of our difficulty estimation verification. To mitigate this limitation, we also considered the GPT4-Turbo ranking as a proxy, but found that GPT4-Turbo's ranking did not closely match human difficulty rankings for problem pairs, indicating it is not a very reliable proxy for human judgment in this context.
- Although we considered a collection of six datasets covering four domains — math, coding, puzzles, and reasoning — this collection might not be exhaustive. This selection does not fully elaborate on all possible domains and tasks that language models could encounter. Other domains and datasets could further enrich and diversify the Easy2Hard-Bench suite.

## D  Dataset Information

### D.1  Legal Compliance

**E2H-AMC.** For the AMC (https://maa.org/math-competitions/amc-8, https://maa.org/math-competitions/amc-1012) and AIME (https://maa.org/math-competitions/american-invitational-mathematics-examination-aime) competitions, the problems are crafted by the Mathematical Association of America (MAA) (https://maa.org/) and the solutions are gathered from the AoPS wiki website. Historically, the MAA has not enforced its intellectual property rights on these problems, even against commercial organizations such as AoPS. This has led to court rulings that the MAA's IP rights have been permanently forfeited. The copyright status of the AoPS wiki (https://artofproblemsolving.com/wiki/index.php/AoPS_Wiki:Copyright) is currently under review, with a notice that states, "Please don't take any non-public domain text from anywhere in the meantime." The problems and solutions for AMC and AIME that we extract are exclusively from publicly accessible pages. The MAA also publishes item difficulty statistics accessible at their website (https://amc-reg.maa.org/reports/generalreports.aspx). For the HMMT competitions, the organizers have not stated any copyright or licensing terms on their official website (https://www.hmmt.org), and all problems and solutions we compile are from openly available PDF and TXT files on their site. In terms of compliance with international copyright laws, we adhere to the Digital Millennium Copyright Act (DMCA) in the United States by not bypassing any access controls. We also ensure compliance with the General Data Protection Regulation (GDPR) in the European Union by anonymizing all identifiers and using the data solely for academic research. Additionally, we only collect a subset of the available problems and their corresponding solutions from all mentioned sources.

**E2H-Codeforces.** We collect the problem text, submission source, and test cases from the publicly accessible pages of Codeforces (https://codeforces.com/). Our practices align with Fair Use § 107, which permits "the fair use of a copyrighted work, including such use by scholarship, or research, is not an infringement of copyright". This is assessed based on "the purpose and character of the use, including whether such use is of a commercial nature or is for nonprofit educational purposes", "the amount and substantiality of the portion used in relation to the copyrighted work as a whole", and "the effect of the use upon the potential market for or value of the copyrighted work." Our dataset, Easy2Hard-Bench, is noncommercial and does not impact the market value of the original problems. Concerning international copyright laws, we adhere to the Digital Millennium Copyright Act (DMCA) in the U.S. and the General Data Protection Regulation (GDPR) in the E.U., with additional details outlined in the AMC section of our documentation.

**E2H-Lichess** The Lichess team states in their Terms of Service (https://lichess.org/terms-of-service) that "Lichess is free/libre open source software. This means that in addition to using our website and its features, technologies, or software for free (collectively referred to as the 'services'), you can also inspect, copy, and (subject to certain licensing requirements) utilize our source code." Furthermore, we either directly access the Lichess open database (https://database.lichess.org/#puzzles) or scrape chess puzzle data from the public pages.

**E2H-GSM8K, E2H-ARC and E2H-Winogrande.** For the three existing datasets, we adhere to their respective licenses. GSM8K is available under the MIT License (https://huggingface.co/datasets/openai/gsm8k#licensing-information), ARC is licensed under CC BY-SA 4.0 (https://huggingface.co/datasets/allenai/ai2_arc), and Winogrande is distributed with a CC-BY 4.0 license (https://github.com/allenai/winogrande?tab=readme-ov-file#license).

### D.2 Author Statement and License

We assume full responsibility for any violations of rights. The Easy2Hard-Bench datasets are licensed under CC BY-SA 4.0 (https://creativecommons.org/licenses/by-sa/4.0/deed.en), while our open-source evaluation code/package is distributed under the Apache License 2.0 (https://apache.org/licenses/LICENSE-2.0).

### D.3 Potential Social Impacts

The problems and solutions in each dataset of Easy2Hard-Bench have been published on the Internet or open source in the previous dataset, indicating that they will not cause any further issues. To clarify, we intend for other researchers to use this dataset to train the models to perform better generalization over difficulty rather than assisting students or players cheat on exams or games.

## E  Dataset Preprocessing Details

Column names:

- **E2H-AMC:** puzzle_id, rating, rating_std, rating_quantile, tag, fen, pgn, annotated_pgn, uci_seq, san_seq, answer_san, answer_uci, init_num_moves, player, popularity_score, puzzle_num_plays, motif_tags, phase_tags, mate_tags, special_move_tags, game_origin_tags, opening_tags, game_hash, game_url, game_pgn, game_annotated_pgn, unnorm_rating, unnorm_rating_std, previous_fen, last_move_uci, problem_text, answer_text, problem_tokens, answer_tokens
- **E2H-Codeforces:** contest, rating, rating_std, rating_quantile, tag, subtest, year, month, index, problem, answer, solution, rating_tag, test_tag, item_difficulty, unnorm_rating, unnorm_rating_std, unnorm_rating_lower, unnorm_rating_upper, ever_exist, problem_text, answer_text, problem_tokens, answer_tokens
- **E2H-Lichess:** contest_id, problem_index, rating, rating_std, rating_volatility, rating_quantile, tag, detailed_tag, problem_name, problem_main, problem_note, input_spec, output_spec, sample_inputs, sample_outputs, inputs, answers, input_output, solution_id_0, solution_0, outputs_0, solution_id_1, solution_1, outputs_1, solution_id_2, solution_2, outputs_2, unnorm_rating, unnorm_rating_std, unnorm_rating_volatility, reference_rating, original_tags, ever_exist, problem_text, answer_text, problem_tokens, answer_tokens

Tags:

- **E2H-AMC:** Hanging Piece, Kingside Attack, Advanced Pawn, Pin, Defensive Move, Discovered Attack, Queenside Attack, Attacking f2 or f7, Skewer, Double Check, Endgame, Rook Endgame, Middlegame, Opening, Knight Endgame, Queen and Rook Endgame, Pawn Endgame, Queen Endgame, Bishop Endgame, Mate in 1, Hook Mate, Double Bishop Mate, Back Rank Mate, Anastasia's Mate, Dovetail Mate, Smothered Mate, En Passant, Promotion, Master Games
- **E2H-Codeforces:** AMC12 First Half, AMC10 Second Half, AMC12 Final Problems, AMC8 Second Half, HMMT Nov Easy, HMMT Feb Easy, HMMT Feb Guts, Hard AIME Problems,

AMC10 Final Problems, AMC8 First Half, AMC12 Second Half, Very Hard AIME Problems, HMMT Feb Team, HMMT Nov Guts, AMC10 First Half, HMMT Nov Hard, HMMT Feb Hard, HMMT Nov Team, Intermediate AIME Problems, Easy AIME Problems, AMC12 A, AMC10 B, AMC12 B, AMC8, HMMT-Nov Theme, HMMT-Feb Combinatorics, HMMT-Feb Guts, AIME, AMC10 A, HMMT-Feb Team, HMMT-Feb Algebra, HMMT-Nov Guts, HMMT-Nov General, HMMT-Feb General, HMMT-Nov Team, HMMT-Feb Calculus, HMMT-Feb Geometry

- **E2H-Lichess:** brute force, sortings, strings, fft, *special, combinatorics, two pointers, geometry, constructive algorithms, trees, math, number theory, data structures, flows, 2-sat, binary search, matrices, graph matchings, implementation, bitmasks, greedy, probabilities, interactive, shortest paths, graphs, games, dsu, hashing, dfs and similar, ternary search, meet-in-the-middle, divide and conquer, string suffix structures, expression parsing, schedules, brute force, greedy, sortings, constructive algorithms, strings, bitmasks, fft, math, number theory, implementation, combinatorics, dp, binary search, data structures, two pointers, geometry, dfs and similar, trees, flows, 2-sat, dsu, graphs, matrices, graph matchings, probabilities, interactive, shortest paths, games, hashing, divide and conquer, ternary search, meet-in-the-middle, string suffix structures, expression parsing, schedules

## E.1 Dataset source

**Mathematics.** For mathematics reasoning, we focus on high-school level mathematics competitions in U.S., such as American Mathematics Competition (AMC), American Invitational Mathematics Examination (AIME), and Harvard-MIT Mathematics Tounrament (HMMT). We choose the problems from these three series among multiple competitions mainly because of the following reasons:

- **Accessibility of problems and solutions.** The problems and solutions are officially published on the Internet and easy for us to collect. Moreover, there are some solutions for problems in AMC and AIME provided by the expert-level users on AoPS, enabling us to improve the quality of solutions for these problems.
- **Reliable statistics of human performance.** Besides problems and solutions, item difficulties for each problems are also provided directly or can be calculated by per-participant results in the official reports. The IRT models are fitted to these human statistics, and we use the corresponding
- **Widely-accepted estimation of competition difficulty level.** The competition ratings by AoPS wiki, which is widely accepted by expert-level users, assigns the competitions approximate difficulty ratings on a scale of 1 to 10. Using these level as a reference standard, we can estimate the unified difficulty scores of the problems across different problems.
- **Broad range of difficulty.** These three series of competitions almost cover all levels of high school mathematics. HMMT February Tournament, the most difficult one, reach the level of Olympiad competitions and is mor difficult than the previous mathematics dataset.
- **QA-friendly formats.** The problems from these competitions are in the format of multiple choice or blank filling. They can be easily adapted for our QA task for LLMs.

**Programming.** For programming, we focus on the online coding contests on Codeforces. We choose the problems from this website because of the following reasons:

- **Accessibility of problems, solutions and testcases.** The problems and massive high quality solutions can be collected from the web page. More importantly, the testcases for each submission by contestants are also available to retrieve from the HTML page.
- **Detailed submission records and contestants rating history.** The submission records of contestants on problems can be downloaded via the API of Codeforces. Moreover, the history of each contestant's rating across the contests is also available via the API. The rating of contestants vary based on their performance in the last participating contest.
- **Granular difficulty score of each problem.** Each problems are labeled with a granular difficulty score by the contest organizers. Although these scores are not continuous-valued metrics, we use them for the sanity check of our difficulty estimation.

**Puzzle solving.** For puzzle solving, we focus on the chess puzzles on Lichess. We choose chess puzzles and Lichess because of the following reasons:

- **Public available human statistics.** Lichess.org is one of the largest online chess platforms, which not only open-sources the code but also publicizes the almost complete game history, evaluation, and puzzle database at https://database.lichess.org/. For other types of puzzles or games

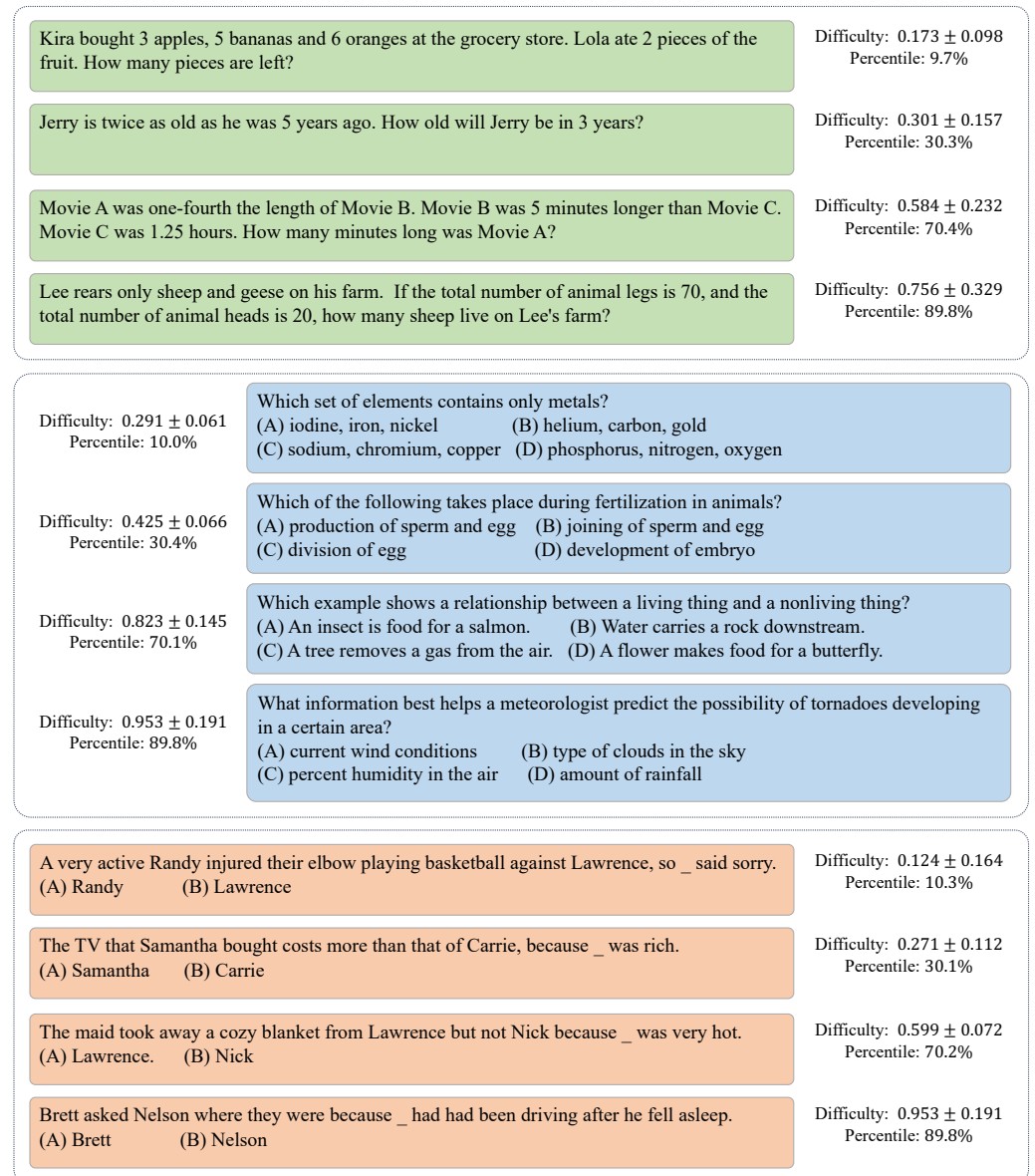

Figure 11: Example problems at different difficulty levels. We present problems from **E2H-GSM8K**, **E2H-ARC**, and **E2H-Winogrande** datasets, illustrating varying difficulty levels within each domain. Higher estimated difficulties correspond to more complex problems, as verified by human studies.

like Go, maze, and sudoku, we cannot find a similar fully publicized platform with such a large user base.

• **Huge amount of puzzles.** On Lichess.org, the chess puzzles are automatically curated using a tiny fraction of selected chess games and Stockfish [47] chess engine evaluations. Because of this, the total number of chess puzzles is large, and the quality of puzzles is also guaranteed.

### E.2 Dataset filtering

For the problems from the aforementioned sources, we filter out some which are not proper for our dataset.

**E2H-AMC.** We exclude the problems satisfying any one of the following conditions: (1) The format of the problem is not friendly for the adaptation to QA task, such as the proof problems without a short answer in HMMT. (2) The inherently multiple-choice problems lose context without the options. (3) There are some external image sources used in the narration of problem, and removing them will cause the problem ill-defined. (4) The problem has more than one correct answer. (5) The correct answer of the problem is not equal to any numerical value, such as a string.

**E2H-Codeforces.** We exclude the problems satisfying any one of the following conditions: (1) The problem does not have any accepted solution in Python. (2) The problem does not have any complete testcase with the corresponding untruncated input and output.

**E2H-Lichess.** We exclude the problems requiring an answer of multiple-moves. These problem can be only adapted to multi-run QA with more complicated metrics.

## E.3 Dataset preprocessing

**E2H-AMC.** We follow the following steps for postprocessing:
1. For AMC and AIME, we collect the problems and high-quality solutions by online users from HTML data on AoPS website. For HMMT, we use the OCR tool Mathpix to obtain LATEXrendered problems and solutions from official materials in PDF documents.
2. We transfer the retrieved HTML source into LATEXrendered text.
3. For the problems in AMC, we convert the format from multiple choice to black filling by removing the options.
4. We remove the personal information of contributors from the solutions.
5. We make that every solution have exactly one corrected answer labeled by "□" by removing the redundant ones or adding one to the end of those solution without it.

**E2H-Codeforces.** We follow the following steps for postprocessing:
1. For each problem, we collect the HTML source from the corresponding web page, retrieve the related paragraphs and convert them into LATEXrendered text.
2. For each problem, we try to select three accepted submissions in Python. If there are more than three accepted one, we choose the ones with the shortest runtime among them.
3. For the collected submissions of each problem, we merge their untruncated testcases and use the union as the testcase for this problem.
4. We remove all comments which could leak the contestants' personal information from the source code of solution.

**E2H-Lichess.** We follow the following steps for postprocessing:
1. We download the files containing information of chess puzzles from Lichess website. In the files the move sequences from the start to the puzzle step is recorded in Forsyth–Edwards Notation (FEN).
2. We convert the move sequence from FEN to Portable Game Notation (PGN) and Universal Chess Interface (UCI) notations by utilizing chess engines.
3. We evaluated the puzzle with Stockfish Chess Engine and collect the result as additional information.

# F Difficulty Verification Details

## F.1 Natural Verification of E2H-AMC, E2H-Codeforces, E2H-Lichess

For the problems in E2H-AMC, E2H-Codeforces and E2H-Lichess, the statistics used for difficulty estimation are human performance metrics in real-world competitions, contests and games. These sources are either authoritative (competition organization committee) or rigorously examined by the professional community. Moreover, both IRT and Glicko-2 are prevalent rating systems used in various scenarios. Thus, we are confident with the difficulty estimation results on these three datasets. Instead of having a large scale of human verification, we randomly sample some problems and check their contents and estimated difficulty. Generally, the estimated results are well aligned with the human understanding to the problems via human's perspective. We use the problems presented in fig. 1 to briefly illustrate the alignment.

**E2H-AMC.** Four problems are from AMC 10 (AoPS rating: 1-2), AMC 12 (AoPS rating:1.5-2), HMMT November (AoPS rating:3.5-5.25), HMMT February (AoPS rating:5.5-6) respectively. The problem from AMC10 is about pre-high-school stage arithmetic and the estimated difficulty median is 0.134. The problem from AMC12 requires some basic knowledge about analytic geometry and the estimated difficulty median is 0.262. The problem from HMMT November is a combinatorics problem requiring some knowledge about number theory, and its estimated difficulty median is 0.587. The problem from HMMT February is a much more challenging combinatorics problem, and its estimated difficulty median is 0.784. Thus, our estimated difficulties are consistent human analysis on these four problems.

**E2H-Codeforces.** Four problems are with four different tags: implement, greedy, math and others. The problem with tag implement can be solved by the combination of basic arithmetic operations, and the estimated difficulty median is 0.134. The problem with tag greedy can be solved by greedy search, and the estimated difficulty median is 0.204. The problem with tag math requires some combinatorics knowledge, and the estimated difficulty median is 0.435. The problem with tag others is a complicated one related to graph theory, and the estimated difficulty median is 0.583. According to the necessary knowledge and skill for problem solving, our estimated difficulties are consistent human analysis on these four problems.

**E2H-Lichess.** Four problems are with four different tags: checkmate, crushing, advantage, equality. In the problem with tag checkmate, the white player can win the game with one-step search. The estimated difficulty median is 0.072. In the problem with tag crushing, the black knight can deliver a family fork at the next step and then gain a queen. The estimated difficulty median is 0.163. In the problem with tag advantage, the white player can capture the black knight with a pawn at the next step. The estimated difficulty median is 0.299. In the problem with tag equality, the answer is using the black knight to exchange white knight, which is actually not the optimal if only considering one step. The estimated difficulty median is 0.418. Based on the advantage and search steps for the next move, our estimated difficulties are consistent human analysis on these four problems.

To sum up, our analysis shows good alignment of our estimated difficulty on these datasets, which is based on high-quality human performance statistics and the rating standard accepted by the expert-level community. Even if more human expert involved in rating, they will mostly agree with our current estimation.

## F.2 Human Difficulty Ranking

For the problems in the datasets E2H-GSM8K, E2H-ARC and E2H-Winogrande, we estimate their difficulty based on the performance metrics of LLMs because there is no accessible human performance records on these problems. That follows a natural question: *are these model performance metrics a good surrogate of human?* To answer this, we design a human difficulty ranking survey for verification.

**Participants.** We recruit the participants from the undergraduate and graduate students. ARC is a dataset of high-school level natural science QA, so all of our participants have enough knowledge to solve these problems. We spread the invitation via email and social media groups to hire the participants. As an incentive, we randomly choose some participants and provide them with 5 dollars as bonus.

**Questionnaire.** The questionnaire is spilt into three parts: (1) Introduction: We briefly introduce the participants' task, determining which question in each pair is more difficult, and the content of three datasets. We illustrate the difficulty as how likely someone with a K-12/12th-grade education level could answer it successfully, and emphasize that difficult questions requires complex computations (GSM8K), more advanced knowledge (ARC). or contain ambiguous or misleading elements (Winogrande). (2) Main body: We order the section of datasets as GSM8K, ARC and Winogrande. In each section, we present 10 pairs of problems from the corresponding dataset. The problems from each pair are sample randomly from the dataset, and the discrepancy of their average accuracy on Open LLM Leaderboard are greater than 0.1. We emphasize that the participants do not need to actually solve the problems,a and encourage them to make selection based on the intuition when they are not so sure. (3) Feedback: At last, we request the participants to provide their feedback on the survey. These feedbacks will be considered in the analysis of survey.

**Sample Size.** We prepare 10 questionnaires with unique problem sets. For each questionnaire, after filtering out invalid results, we receive the responses from at least five different participants. Therefore, we collect the responses from 50 participants on 100 problems from each dataset. We use the majority vote of responses on each problem pair as the result of human verification.

**Institutional Review Board (IRB).** Prior to the start of this human verification survey, we report all experimental setup and design to Division of Research, University of Maryland for institutional review board. The survey is determined as exempt form IRB review according to federal regulations.

---

**GSM8K**

> 1. Maddison has 5 boxes with 50 marbles in each box. Then she gets 20 marbles from her friend. How many marbles does she have now?
> 2. Seth gave half of his stickers to Luis. Luis used half of the stickers and gave the rest to Kris. Kris kept 9 of the stickers and gave the remaining 7 stickers to Rob. How many stickers did Seth have in the beginning?

IRT estimation (difficulty, quantile): $(0.190 \pm 0.111, 12.4\%)$, $(0.759 \pm 0.295, 90.1\%)$

Human Preference (harder problem): 2, 2, 2, 2, 2

GPT4 scores: (3.0, 5.0), (3.0, 6.0), (3.0, 6.0)

**ARC**

> 1. If you place a thermometer into a glass of ice water, what temperature should the thermometer read?
>    (A) -10°C        (B) 0°C        (C) 32°C        (D) 100°C
> 2. Water evaporation on the surface of Earth most likely causes the formation of
>    (A) glaciers        (B) mountains        (C) natural gas        (D) limestone.

IRT estimation (difficulty, quantile): $(0.524 \pm 0.262, 44.8\%)$, $(0.972 \pm 0.170, 93.8\%)$

Human Preference (harder problem): 2, 1, 2, 2, 2

GPT4 scores: (2.0, 6.0), (2.0, 6.0), (2.0, 6.0)

**Winogrande**

> 1. Cynthia felt very thirsty but Sarah did not feel thirsty.
>    (A) Cynthia bought a bag of chips.        (B) Sarah bought a bag of chips.
> 2. The snow came down so much that Michael had to go plow Kevins driveway because
>    (A) Michael needed the help of his neighbors.        (B) Kevin needed the help of his neighbors.

IRT estimation (difficulty, quantile): $(0.080 \pm 0.186, 6.0\%)$, $(0.942 \pm 0.109, 90.1\%)$

Human Preference (harder problem): 2, 2, 1, 2, 2

GPT4 scores: (4.0, 5.0), (4.0, 3.0), (6.0, 3.0)

---

### F.3 Compare with Human

We use the majority vote from 5 participants on each problem as the human preference. During the computation of matching accuracy and average per-pair discrepancy, we ignore those pairs where the discrepancy of two problems' IRT difficulty in a pair is less than the maximum IRT difficulty standard deviation between the two problems.

For the results shown in table 3, the alignments in ARC and Winogrande are not as satisfying as GSM8K. We mention here that it may not be blamed on the IRT method. In the human verification survey, we noticed some participants' feedback complaining the problems in ARC and Winogrande are harder to rank their difficulty. They comment like "To me, It's hard to compare the question pairs in the second (ARC) and third (Winogrande) tasks" or "The final section (Winogrande) was more challenging. I realized that the more "unclear" the answer could be made the question more

Table 4: Verification of estimated difficulties on E2H-GSM8K, E2H-ARC, and E2H-Winogrande, which are based on collective statistics of LLMs and obtained using Item Response Theory (IRT). IRT-estimated difficulties align well with human preferences and outperform the alignment with GPT4.

| | Metric | GSM8K | ARC | Winogrande |
|---|---|---|---|---|
| GPT4 v.s. Human | Matching Acc. | 0.922 | 0.825 | 0.771 |
| | Avg. Per-pair Discrepancy | 0.029 | 0.055 | 0.065 |
| IRT v.s. GPT4 | Spearman Corr. | 0.612 | 0.218 | 0.164 |

difficult". These human feedbacks show that the difficulties in these two datasets are more vague even for human subjects.

### F.4  GPT4 Difficulty Ranking

Although we initially evaluate IRT-based difficulty estimation with human evaluation as the standard, the limited pool of participants makes the scaling-up of verification infeasible. Considering high reasoning ability of state-of-art LLMs, we use GPT4-Turbo as a proxy to scale up the number of problem pairs in verification. For each dataset, we rank 2000 pairs of problems with the specific prompt. Different from human evaluation, we requested GPT4 in the prompts to score the difficulty of both problems with integers from 1 to 10 rather than merely ranking two problems. We sample three times for each problem and compare the average scores in each pair. In the prompt for each dataset, we list several factor to consider in the evaluation. See the specific prompts in appendix J.

### F.5  Compare with GPT4

#### F.5.1  Compare GPT4 with Human

To verify that GPT4 is a reliable surrogate of humans, we compare the GPT4 verification results with human verification results. Similar to the comparison between IRT and humans, we exclude the pairs with an average GPT score discrepancy not greater than 2.0. The results in table 4 show that GPT-4 also achieves similar behavior with human participants in difficulty ranking. This validates the scaling-up by GPT4 as a surrogate of humans.

#### F.5.2  Compare GPT4 with IRT

For the comparison between the IRT method and GPT4 rankings, we exclude those with an IRT discrepancy less than the maximum IRT standard deviation and those with a GPT discrepancy not greater than 2.0. In other words, we keep the same criterion in IRT v.s. human and GPT4 v.s. human.

Besides matching accuracy and average per-pair discrepancy, we report Spearman correlations in table 4. All results are unsatisfying. Although both GPT4 and IRT show a relatively good alignment with human verification, they cannot align well with each other. Although the GPT4 rankings are not well correlated with the IRT difficulty rankings, shown by the Spearman correlation in table 4, it only indicates that GPT4 may not be good proxy on ranking difficulty of pairs of problems. The alignment metrics between IRT and Human ranking validated in table 3 in section 2 already justify the well alignment of IRT estimated difficulty and the human consensus of difficulty.

## G  Details on Benchmarking Performance

This section provides comprehensive details on the benchmarking performances of various large language models (LLMs) evaluated using the Easy2Hard-Bench. We cover the details on model selection, evaluation setups and metrics.

## G.1 Model Selections and Details

We evaluated a range of state-of-the-art LLMs from both proprietary and open-source families to understand their capabilities in solving increasingly difficult problems across different domains. For the OpenAI model, we utilized the Azure OpenAI platform. Claude and Gemini models were accessed through their respective official APIs. For the open-sourced models Llama, Mixtral, and Qwen, we employed the self-hosted LLM API with their provided quantized models.

**Model details:**
- **GPT-4-Turbo** (gpt-4-2025-04-23), accessed via `https://learn.microsoft.com/en-us/azure/ai-services/openai/concepts/models#gpt-4-turbo`
- **Claude3-Opus** (claude-3-opus-20240229), accessed via `https://docs.anthropic.com/en/docs/models-overview#claude-3-a-new-generation-of-ai`
- **Gemini-1.5-Pro** (gemini-1.5-pro-latest), accessed via `https://ai.google.dev/gemini-api/docs/models/gemini`
- **Llama3-70B** (llama3:70b-instruct-q5_K_M), accessed via `https://ollama.com/library/llama3:70b-instruct-q5_K_M`
- **Mixtral-8x22B** (mixtral:8x22b-instruct-v0.1-q5_K_M), accessed via `https://ollama.com/library/mixtral:8x22b-instruct-v0.1-q5_K_M`
- **Qwen1.5-110B** (qwen:110b-chat-v1.5-q5_K_M), accessed via `https://ollama.com/library/qwen:110b-chat-v1.5-q5_K_M`

## G.2 Evaluation Setups and Metrics

This subsection provides in-depth insights into the metrics and evaluation setups used for assessing the performance of state-of-the-art LLMs on our Easy2Hard-Bench. Our approach adapts established benchmarks to the unique challenges presented by specific tasks, ensuring a robust and fair evaluation of each model's capabilities.

**Detailed Evaluation Metrics:**
- **E2H-AMC:** For mathematical problems, we require the solutions to be submitted in LaTeX format, specifically enclosed within "□". To measure accuracy, we match the correct answers within these boxes. During the preprocessing phase, we ensure that all responses in the AMC dataset are parsable by the sympy latex parsing function, available at `https://docs.sympy.org/latest/modules/parsing.html#experimental-mathrm-latex-parsing`. This approach compares the parsed sympy expression with the provided answer expression. It effectively addresses variations in the way latex might typeset equations, such as differences in spacing between terms. Solutions that contain unparsable latex equations are automatically marked as incorrect.
- **E2H-Codeforces:** In our assessment of programming challenges, the primary metric is the test case average accuracy, following the model used in the APPS benchmark. We also support additional metrics like strict accuracy and pass@k, as described in the HumanEval paper. The test case average accuracy is computed using the formula:

$$\frac{1}{P} \sum_{p=1}^{P} \frac{1}{C_p} \sum_{c=1}^{C_p} \mathbb{1} \left\{ \text{eval} \left( \langle \text{code}_p \rangle, x_{p,c} \right) = y_{p,c} \right\}.$$

  This measurement evaluates the average fraction of test cases that each submitted solution passes for a given problem. It allows us to assess partial successes and pinpoint areas where models may need improvement, as it is common for solutions to pass some test cases but fail others.
- **E2H-Lichess:** For the evaluation of chess puzzles, we have developed a unique QA format. Each chess puzzle is provided to the LLMs in four different formats to ensure comprehensive understanding: (1) the FEN notation of the current board configuration, (2) a non-annotated PGN notation tracking all moves from the start of the game to the current position, (3) an annotated PGN notation provided by the Stockfish chess engine that includes evaluations and win-rate predictions for each move, and (4) the UCI notation for previous moves. The models are prompted to predict the next best move in both UCI and PGN notations. A successful match in either notation against the expected move is considered a correct solution. Full details of the prompt template can be found in Section J of this document.
- **Existing Datasets (E2H-GSM8K, E2H-ARC, E2H-Winogrande):** For these datasets, we adhere strictly to the protocols established by the Open LLM Leaderboard, utilizing the llm-evaluation-

harness package provided by EleutherAI, available at `https://github.com/EleutherAI/lm-evaluation-harness`. This standardized evaluation framework ensures that our model performance assessments are consistent and comparable with other leading benchmarks in the field.

## H  Details on Profiling Generalization

This section provides comprehensive details on the profiling the easy-to-hard generalizations of various large language models (LLMs) using the Easy2Hard-Bench. We cover the details on experiment setups, post-processing and visualization.

### H.1  Experiment Setups

The following details expand on the experimental setups described in the main text, providing specific configurations and adjustments made for each dataset.

**E2H-AMC and E2H-GSM8K:** The experiments conducted on the AMC and GSM8K datasets were facilitated using specific model checkpoints and training settings:

- The GPT3.5-Turbo models used in our experiments correspond to the gpt-35-turbo-0613 checkpoints, deployed on Azure OpenAI services. These were finetuned adhering to the default training parameters, spanning 3 epochs with a learning rate factor of 1.
- For E2H-AMC, the roles of the training and evaluation splits were switched specifically for this experiment. The complete training split now encompasses 2,975 samples, hence each training bin contains around 372 samples (i.e., $2975/(7 + 1)$).
- In E2H-GSM8K, we initially randomly sampled the evaluation split with 359 samples, leaving the remaining for training. Thus, each training bin consists of 120 samples (i.e., $(1319 - 359)/(7+1)$).
- Despite the training set size being relatively small, such as only 120 samples, it aligns well with OpenAI's finetuning guide which suggests, "To fine-tune a model, you are required to provide at least 10 examples. We typically see clear improvements from fine-tuning on 50 to 100 training examples with gpt-3.5-turbo but the right number varies greatly based on the exact use case." (`https://platform.openai.com/docs/guides/fine-tuning/example-count-recommendations`)

**E2H-Lichess:** For the E2H-Lichess dataset, a specialized approach involving tailored tokenization and model selection was implemented to enhance the performance:

- The tokenizer employed for this dataset was based on the UCI notation, simplistically designed to encode chess moves efficiently. It tokenizes each chess move into three tokens: one for the chess piece moved (six possible pieces), one for the starting position, and one for the ending position on the board (64 possibilities each). Additionally, the game result is denoted by one of three tokens: win, loss, or tie.
- The model used, Leon-Chess-1M-BOS, was specifically trained on a corpus of 1 million real-world chess games sourced from Lichess, distinct from the puzzles used in Easy2Hard-Bench, ensuring no overlap and thus preserving the integrity of the dataset. The model is publicly available at `https://huggingface.co/Leon-LLM/Leon-Chess-1M-BOS`.
- During training and evaluation on the chess puzzles in Easy2Hard-Bench, only the completion of the move (i.e., the next move in a puzzle) was considered for the SFT loss. The evaluation metric was strict, considering a prediction correct only if all three tokens representing the next move matched perfectly.

### H.2  Post-processing and Visualization

This section elucidates the methodologies used for postprocessing and visualizing the interpolated results of the generation gain across various training difficulties.

**Interpolation of Results:** As noted in the main text, our visualizations (counter/heatmap) display the interpolated generation gain from arbitrary continuous training difficulties to arbitrary continuous evaluation difficulties. This requires interpolation between results on 7 discrete training difficulties to generate a continuous map. We utilize the robust Radial Basis Function (RBF) kernel interpolation algorithm via `scipy.interpolate.RBFInterpolator` (`https://docs.scipy.org/doc/scipy/`

`reference/generated/scipy.interpolate.RBFInterpolator.html`), ensuring no deviation from actual observations. The smoothing factor is set to 0.0, indicating exact interpolation without smoothing, where the interpolated function strictly passes through the nodal points. This guarantees that our visualizations accurately reflect the actual results on the 7 discrete training difficulties. The kernel used for this interpolation is cubic.

**Reduction of Randomness:** To address potential randomness and improve the robustness of the results from training bins with random difficulty, we conduct the training process twice for the random difficulty bin and average the performance metrics. This method helps in stabilizing the background generation behavior, thus providing more reliable insights into the effects of varied training intensities.

# I  Dataset Sheet

This appendix presents a datasheet for the Easy2Hard-Bench dataset. We use the format in [18] for our datasheet.

## I.1  Motivation

1. **For what purpose was the dataset created?** *Was there a specific task in mind? Was there a specific gap that needed to be filled? Please provide a description.*

   Current benchmarks cannot predominantly measure the easy to hard progression systematically, which is essential for applications in curriculum learning and understanding the evolution of AI from simpler to more advanced problem-solving abilities. While numerous datasets exist, they often fall short of providing a structured framework that mirrors the gradual complexity increase encountered in real-world scenarios, crucial for effectively benchmarking and enhancing the adaptability and learning curriculum of LLMs. Filling this gap is the main purpose of creating this dataset.

2. **Who created this dataset (e.g., which team, research group) and on behalf of which entity (e.g., company, institution, organization)?**

   The authors of this paper created Easy2Hard-Bench. The core members are from Prof. Furong Huang's research group at the University of Maryland.

3. **Who funded the creation of the dataset?** *If there is an associated grant, please provide the name of the grantor and the grant name and number.*

   The creation of the dataset is supported by DARPA Transfer from Imprecise and Abstract Models to Autonomous Technologies (TIAMAT) 80321, National Science Foundation NSF-IIS-2147276 FAI, DOD-ONR-Office of Naval Research under award number N00014-22-1-2335, DOD-AFOSR-Air Force Office of Scientific Research under award number FA9550-23-1-0048, DOD-DARPA-Defense Advanced Research Projects Agency Guaranteeing AI Robustness against Deception (GARD) HR00112020007, Adobe, Capital One and JP Morgan faculty fellowships.

4. **Any other comments?**

   No.

## I.2  Composition

1. **What do the instances that comprise the dataset represent (e.g., documents, photos, people, countries)?** *Are there multiple types of instances (e.g., movies, users, and ratings; people and interactions between them; nodes and edges)? Please provide a description.*

   Easy2Hard-Bench consists of six datasets spanning six distinct domains, including mathematics problem solving, competitive programming, chess puzzles, and various common-sense reasoning tasks. Each instance of a dataset represent a question, i.e., a problem able to solve, state-able in natural language or clear textual notations (e.g., math equations, programming languages, and chess notations), and each associate with a unique answer (or a characterizable set of answers) also in natural language.

2. **How many instances are there in total (of each type, if appropriate)?**

   There are 3975 problems in E2H-AMC, 7663 problems in E2H-Codeforces, 76763 problems in E2H-Lichess, 1319 in E2H-GSM8K, 1172 in E2H-ARC, 1267 in E2H-Winogrande.

3. **Does the dataset contain all possible instances or is it a sample (not necessarily random) of instances from a larger set?** *If the dataset is a sample, then what is the larger set? Is the sample representative of the larger set (e.g., geographic coverage)? If so, please describe how this representativeness was validated/verified. If it is not representative of the larger set, please describe why not (e.g., to cover a more diverse range of instances, because instances were withheld or unavailable).*

   It is not feasible to represent all mathematical fields across all dimensions of "mathematical behavior" and all types of mathematical questions

4. **What data does each instance consist of? "Raw" data (e.g., unprocessed text or images) or features?** *In either case, please provide a description.*

   The text of problem, answer, and solution (if applicable). The numerical value of difficulty. And the tags in the form of strings. Each problem from these datasets at least consists of the textual prompt (or equivalently, textural information fields provided in the prompt), and the corresponding answers in natural language. From E2H-AMC, E2H-Codeforces, and E2H-Lichess datasets, fine-grained, multiple types of categorical tags are provided. For E2H-AMC, each problem is also associated with a textual solution. For E2H-Codeforces, test cases in form of inputs and expected outputs are list of texts are also included.

5. **Is there a label or target associated with each instance?** *If so, please provide a description.*

   Yes, each instance is associated with the ground-truth answer or test cases providing correctness.

6. **Is any information missing from individual instances?** *If so, please provide a description, explaining why this information is missing (e.g., because it was unavailable). This does not include intentionally removed information, but might include, e.g., redacted text.*

   No.

7. **Are relationships between individual instances made explicit (e.g., users' movie ratings, social network links)?** *If so, please describe how these relationships are made explicit.*

   We removed duplicate problems in each dataset.

8. **Are there recommended data splits (e.g., training, development/validation, testing)?** *If so, please provide a description of these splits, explaining the rationale behind them.*

   We split each dataset into training and evaluation datasets. The specific size of splits in each dataset are reported in table 2. We split the dataset mainly based on the number of problems. We aim to guarantee a evaluation split of size larger than 2500 but less than 5000 problems. The size of the evaluation split should be large enough for low granularity of difficulty. But it also should be not too huge considering the cost of evaluation.

9. **Are there any errors, sources of noise, or redundancies in the dataset?** *If so, please provide a description.*

   See appendix E.

10. **Is the dataset self-contained, or does it link to or otherwise rely on external resources (e.g., websites, tweets, other datasets)?** *If it links to or relies on external resources, a) are there guarantees that they will exist, and remain constant, over time; b) are there official archival versions of the complete dataset (i.e., including the external resources as they existed at the time the dataset was created); c) are there any restrictions (e.g., licenses, fees) associated with any of the external resources that might apply to a future user? Please provide descriptions of all external resources and any restrictions associated with them, as well as links or other access points, as appropriate.*

    The dataset is self-contained.

11. **Does the dataset contain data that might be considered confidential (e.g., data that is protected by legal privilege or by doctor-patient confidentiality, data that includes the content of individuals non-public communications)?** *If so, please provide a description.*

    No.

12. **Does the dataset contain data that, if viewed directly, might be offensive, insulting, threatening, or might otherwise cause anxiety?** *If so, please describe why.*

    No.

13. **Does the dataset relate to people?** *If not, you may skip the remaining questions in this section.*

    Yes.

14. **Does the dataset identify any subpopulations (e.g., by age, gender)?** *If so, please describe how these subpopulations are identified and provide a description of their respective distributions within the dataset.*

    No.

15. **Is it possible to identify individuals (i.e., one or more natural persons), either directly or indirectly (i.e., in combination with other data) from the dataset?** *If so, please describe how.*

    For the dataset E2H-AMC, although we have tried to clean the contributor's username on AoPS from the solutions of the problems as much as possible, there could be still some left (<1%). The username can be used to identify individuals indirectly.

16. **Does the dataset contain data that might be considered sensitive in any way (e.g., data that reveals racial or ethnic origins, sexual orientations, religious beliefs, political opinions or union memberships, or locations; financial or health data; biometric or genetic data; forms of government identification, such as social security numbers; criminal history)?** *If so, please provide a description.*

    No.

17. **Any other comments?**

    No.

## I.3 Collection Process

1. **How was the data associated with each instance acquired?** *Was the data directly observable (e.g., raw text, movie ratings), reported by subjects (e.g., survey responses), or indirectly inferred/derived from other data (e.g., part-of-speech tags, model-based guesses for age or language)? If data was reported by subjects or indirectly inferred/derived from other data, was the data validated/verified? If so, please describe how.*

    Some data was collected by scraping texts from the corresponding website, while others were retrieved from TXT or PDF (with OCR tools) files. We introduce the process in detail in E.

2. **What mechanisms or procedures were used to collect the data (e.g., hardware apparatus or sensor, manual human curation, software program, software API)?** *How were these mechanisms or procedures validated?*

    We used self-made scrapers based on Python, and we checked the scraped data manually to make sure it matched the source.

3. **If the dataset is a sample from a larger set, what was the sampling strategy (e.g., deterministic, probabilistic with specific sampling probabilities)?**

    Some problems we scraped were left out of E2H-AMC, E2H-Codeforces, and E2H-Lichess for various reasons. We refer the details to appendix E.

4. **Who was involved in the data collection process (e.g., students, crowdworkers, contractors) and how were they compensated (e.g., how much were crowdworkers paid)?**

    The data collection was mainly finished by the authors, and some undergraduate and graduate students were involved in the data collection process. We refer their compensation to appendix F .

5. **Over what timeframe was the data collected? Does this timeframe match the creation timeframe of the data associated with the instances (e.g., recent crawl of old news articles)?** *If not, please describe the timeframe in which the data associated with the instances was created.*

    The was collected from March to May 2024. Generally, the timeframe in which the data associated with the instances was created is from 2000 to 2024.

6. **Were any ethical review processes conducted (e.g., by an institutional review board)?** *If so, please provide a description of these review processes, including the outcomes, as well as a link or other access point to any supporting documentation.*

    An institutional review board (IRB) was conducted by Division of Research, University of Maryland. We report the project and the human evaluation part in detail. The survey is determined as exempt from IRB review according to federal regulations.

7. **Does the dataset relate to people?** *If not, you may skip the remaining questions in this section.*

    Yes.

8. **Did you collect the data from the individuals in question directly, or obtain it via third parties or other sources (e.g., websites)?**

    We collect the data from the individuals directly.

9. **Were the individuals in question notified about the data collection?** *If so, please describe (or show with screenshots or other information) how notice was provided, and provide a link or other access point to, or otherwise reproduce, the exact language of the notification itself.*

   Yes. We introduce the goal of human evaluation at the start of the survey. The questionnaire is presented in appendix J.

10. **Did the individuals in question consent to the collection and use of their data?** *If so, please describe (or show with screenshots or other information) how consent was requested and provided, and provide a link or other access point to, or otherwise reproduce, the exact language to which the individuals consented.*

    N.A.

11. **If consent was obtained, were the consenting individuals provided with a mechanism to revoke their consent in the future or for certain uses?** *If so, please provide a description, as well as a link or other access point to the mechanism (if appropriate).*

    N.A.

12. **Has an analysis of the potential impact of the dataset and its use on data subjects (e.g., a data protection impact analysis) been conducted?** *If so, please provide a description of this analysis, including the outcomes, as well as a link or other access point to any supporting documentation.*

    No.

13. **Any other comments?**

    No.

### I.4 Preprocessing, cleaning and labeling

1. **Was any preprocessing/cleaning/labeling of the data done (e.g., discretization or bucketing, tokenization, part-of-speech tagging, SIFT feature extraction, removal of instances, processing of missing values)?** *If so, please provide a description. If not, you may skip the remainder of the questions in this section.*

   Yes. We describe in appendix E.

2. **Was the "raw" data saved in addition to the preprocessed/cleaned/labeled data (e.g., to support unanticipated future uses)?** *If so, please provide a link or other access point to the "raw" data.*

   No.

3. **Is the software used to preprocess/clean/label the instances available?** *If so, please provide a link or other access point.*

   Not at this time.

4. **Any other comments?**

   No.

### I.5 Uses

1. **Has the dataset been used for any tasks already?** *If so, please provide a description.*

   Yes. See section 3 and section 4.

2. **Is there a repository that links to any or all papers or systems that use the dataset?** *If so, please provide a link or other access point.*

   No.

3. **What (other) tasks could the dataset be used for?**

   N.A.

4. **Is there anything about the composition of the dataset or the way it was collected and preprocessed/cleaned/labeled that might impact future uses?** *For example, is there anything that a future user might need to know to avoid uses that could result in unfair treatment of individuals or groups (e.g., stereotyping, quality of service issues) or other undesirable harms (e.g., financial harms, legal risks) If so, please provide a description. Is there anything a future user could do to mitigate these undesirable harms?*

   We illustrate the legal compliance of data collection in appendix D.

5. **Are there tasks for which the dataset should not be used?** *If so, please provide a description.*
   No.
6. **Any other comments?**
   No.

## I.6 Distribution

1. **Will the dataset be distributed to third parties outside of the entity (e.g., company, institution, organization) on behalf of which the dataset was created?** *If so, please provide a description.*
   Yes. The dataset will be publicly distributed.
2. **How will the dataset will be distributed (e.g., tarball on website, API, GitHub)** *Does the dataset have a digital object identifier (DOI)?*
   The dataset is available at the Hugging Face collection `https://huggingface.co/collections/furonghuang-lab/easy2hard-bench-666a0d26f3932ecb92c112c2`.
3. **When will the dataset be distributed?**
   The dataset is currently available.
4. **Will the dataset be distributed under a copyright or other intellectual property (IP) license, and/or under applicable terms of use (ToU)?** *If so, please describe this license and/or ToU, and provide a link or other access point to, or otherwise reproduce, any relevant licensing terms or ToU, as well as any fees associated with these restrictions.*
   We release the dataset under the following Creative Commons license: Attribution-NonCommercial 4.0 International (CC BY-NC 4.0). See appendix D for more information.
5. **Have any third parties imposed IP-based or other restrictions on the data associated with the instances?** *If so, please describe these restrictions, and provide a link or other access point to, or otherwise reproduce, any relevant licensing terms, as well as any fees associated with these restrictions.*
   The data in E2H-ARC and E2H-Winogrande are licensed under CC BY-SA 4.0. There are also some IP restrictions applying to the source of E2H-AMC, E2H-Codeforces, and E2H-Lichess. We refer the details to appendix D.
6. **Do any export controls or other regulatory restrictions apply to the dataset or to individual instances?** *If so, please describe these restrictions, and provide a link or other access point to, or otherwise reproduce, any supporting documentation.*
   No.
7. **Any other comments?**
   No.

## I.7 Maintenance

1. **Who will be supporting/hosting/maintaining the dataset?**
   The dataset will be hosted as a Hugging Face repository.
2. **How can the owner/curator/manager of the dataset be contacted (e.g., email address)?**
   The email addresses of the correspondence authors are available. Moreover, the authors can be contacted by raising issues on Github or Hugging Face.
3. **Is there an erratum?** *If so, please provide a link or other access point.*
   Not at this time. But we will have one on Hugging Face.
4. **Will the dataset be updated (e.g., to correct labeling errors, add new instances, delete instances)?** *If so, please describe how often, by whom, and how updates will be communicated to users (e.g., mailing list, GitHub)?*
   The authors will add new instances and correct the potential errors. There will be probably two updates by the authors per year, and these changes will be announced in Hugging Face.
5. **If the dataset relates to people, are there applicable limits on the retention of the data associated with the instances (e.g., were individuals in question told that their data would be retained for a fixed period of time and then deleted)?** *If so, please describe these limits and explain how they will be enforced.*
   N.A.

6. **Will older versions of the dataset continue to be supported/hosted/maintained?** *If so, please describe how. If not, please describe how its obsolescence will be communicated to users.*

   Yes. Older versions will be available in the Hugging Face history, and the corresponding commits will be archived in the README file.

7. **If others want to extend/augment/build on/contribute to the dataset, is there a mechanism for them to do so?** *If so, please provide a description. Will these contributions be validated/verified? If so, please describe how. If not, why not? Is there a process for communicating/distributing these contributions to other users? If so, please provide a description.*

   Yes, the dataset can be extended with additional problems with difficulty following the existing format.

8. **Any other comments?**

   No.

## J    Examples and Templates

### J.1    Questionnaire Templates for Human Evaluation

**Introdction**

> Welcome to our test evaluating the difficulty of questions from popular large language model datasets, including GSM8K, ARC, and Winogrande. You will be presented with ten pairs of questions from each of these datasets. Your task is to determine **which question in each pair is more difficult** for the average high school graduate to answer correctly.
>
> We consider a question more difficult if it is less likely that someone with a K-12/12th-grade education level could answer it successfully. Questions that require complex reasoning and computations, more advanced knowledge, or contain ambiguous or misleading elements tend to be harder.
>
> Please answer to the best of your ability. If you have any questions about the survey, please contact xxxxx@xxx.xxx.
>
> This survey should take around 20 minutes to complete.

**GSM8K Information**

> The GSM8K dataset contains math word problems that require comprehension and the application of **arithmetic operations** in real-life contexts. Here is an example of a question and answer:
>
> Question: 15 gallons of gas were equally divided into 5 different containers. Josey needed 1/4 of a container to run her lawnmower. How many pints of gasoline did Josey need?
>
> Answer: 15 gallons = 120 pints. 120/5 = 24 pints per container. (1/4)24 = 6 pints. Josey needed 6 pints of gas for her lawnmower.
>
> **Reminder**: You will not need to solve the problem. Simply choose which problem would be more difficult to solve.

**ARC Information**

> The ARC dataset consists of science-based multiple-choice questions that test **scientific knowledge and reasoning abilities**. Here is an example question and answer:
>
> Question: A 0.20 kg softball travels 97 meters (m) south for 4.5 seconds (s). What piece of information distinguishes the velocity from the speed of the ball?
>
> Choices: (A) The ball went south. (B) The ball flew for 4.5 s. (C) The ball traveled 97 m. (D) The ball has a mass of 0.20 kg.
>
> Correct Answer: (A) The ball went south.

**Winogrande Information**

The Winogrande dataset has **commonsense reasoning** questions about interactions between entities in real-world situations. The questions are presented as minimal pairs differing by one word that changes the answer. Here is an example question and answer:

Question: Aaron didn't know Dennis had a peanut allergy, so when

Choices: (A) Aaron ate peanut chicken an ambulance was called. (B) Dennis ate peanut chicken an ambulance was called.

Correct Choice: (B) Dennis ate peanut chicken an ambulance was called.

**Reminder**: You do not need to know the correct answer. Choose the more difficult question based on how challenging it would be to select the correct answer given the question context and answer choices.

**Question**

Please identify the more challenging question from the following pair. If you encounter problems with similar difficulty and are unsure, please still make a selection based on your intuition.

☐ {Problem 0}

☐ {Problem 1}

**Feedback**

Did you find it difficult to understand the questions in this dataset or to compare the difficulty of the question pairs? If so, please briefly describe what you found challenging or unclear. We appreciate any additional feedback you may have as well.

[                                    ]

### J.2 Pormpt templates for GPT4 ranking

**GSM8K**

```
system_prompt:
```

You are an impartial judge tasked with determining the difficulty level of math word problems, which require comprehension and the application of mathematical operations within real-life contexts.

```
user_prompt:
```

Please assist in evaluating the difficulty of math word problems. The human testers are more likely to struggle with questions that exhibit higher complexity. You should assign a higher difficulty score to the more challenging question.

When evaluating difficulty, consider the following factors derived from both computational complexity and linguistic analysis:
1. **Number of Calculation Steps**: More computational steps generally increase the probability of errors, indicating a higher difficulty level.
2. **Number of Objects**: Questions involving multiple objects typically require more complex logical analysis, thus increasing difficulty.
3. **Quantitative Relationships Among Objects**: The presence of intricate quantitative relationships demands extended reasoning, contributing to higher difficulty.

4. **Numerical Precision and Lexical Difficulty**: The usage of numerically dense language and specific mathematical operations, coupled with high average word rank and readability scores (such as the Flesch-Kincaid grade level), can significantly affect problem complexity.
5. **Diversity and Frequency of Mathematical Operations**: A variety of used operations and their frequency (e.g., addition, division) influence the cognitive load required to solve the problems.
6. **Depth of Linguistic Structure**: Deeper constituency tree depths indicate more complex sentence structures, potentially increasing the cognitive load for problem-solving.
7. **Relevance of World Knowledge**: Problems requiring specific real-world knowledge or contextual information are typically more challenging.

In your assessment, you will be provided with both the questions and their corresponding answers. Use these answers to verify if the problem aligns with the factors mentioned above.

The scoring for each question should range from 1 to 10, where a score above 5 indicates that the question aligns strongly with these complexity indicators, suggesting a higher difficulty. Conversely, a score below 5 suggests a question is relatively less complex and easier to solve.

Start by offering a brief comparative analysis of the two questions based on the above criteria. Then, present your scores in the format: "[[score1, score2]]", where "score1" represents your assigned difficulty score for Question A, and "score2" stands for Question B. Ensure to maintain objectivity, eliminating any positional or length biases in your evaluation.

[The Start of Question A]

{question0}

[The End of Question A]

[The Start of Question A's Answer]

{answer0}

[The End of Question A's Answer]

[The Start of Question B]

{question1}

[The End of Question B]

[The Start of Question B's Answer]

{answer1}

[The End of Question B's Answer]

**ARC**

system_prompt:

You are tasked with evaluating the difficulty of science-based multiple-choice questions, which require varied levels of scientific knowledge and reasoning.

user_prompt:

Please assist in evaluating the difficulty of math word problems. The human testers are more likely to struggle with questions that exhibit higher complexity. You should assign a higher difficulty score to the more challenging question.

When evaluating difficulty, consider the following factors derived from both computational complexity and linguistic analysis:
1. **Complexity of Scientific Concepts**: The presence of advanced or less commonly encountered scientific concepts indicates a higher difficulty level. Such questions test deeper understanding and application of scientific principles.

2. **Specificity of Knowledge Required**: Questions demanding specific knowledge that is not broadly known or intuitive are often more difficult, as they test the limits of the test taker's factual and conceptual science knowledge.
3. **Depth of Required Reasoning**: Questions requiring multilayered reasoning or complex problem-solving skills suggest higher difficulty. They often involve analyzing multiple components or hypothetical scenarios.
4. **Presence of Distractors**: Answer choices that include plausible but incorrect options based on common misconceptions or closely related concepts can make a question more challenging due to their potential to mislead.
5. **Linguistic Complexity**: The use of complex language, specialized vocabulary, or dense question structures can increase cognitive load, making the question harder to comprehend and analyze.

In your assessment, you will be provided with both the questions and their corresponding answers. Use these answers to verify if the problem aligns with the factors mentioned above.

The scoring for each question should range from 1 to 10, where a score above 5 indicates that the question aligns strongly with these complexity indicators, suggesting a higher difficulty. Conversely, a score below 5 suggests a question is relatively less complex and easier to solve.

Start by offering a brief comparative analysis of the two questions based on the above criteria. Then, present your scores in the format: "[[score1, score2]]", where "score1" represents your assigned difficulty score for Question A, and "score2" stands for Question B. Ensure to maintain objectivity, eliminating any positional or length biases in your evaluation.

[The Start of Question A]

{question0}

[The End of Question A]

[The Start of Question A's Choices]

{choices0}

[The End of Question A's Choices]

[The Start of Question A's Correct Choice]

{target0}

[The End of Question A's Correct Choice]

[The Start of Question A]

{question1}

[The End of Question A]

[The Start of Question A's Choices]

{choices1}

[The End of Question A's Choices]

[The Start of Question A's Correct Choice]

{target1}

[The End of Question B's Correct Choice]

**Winogrande**

```
system_prompt:
```

You are an impartial judge tasked with determining the difficulty level of commonsense reasoning questions that require an understanding of interactions between entities grounded in real-world situations.

```
user_prompt:
```

Please assist in evaluating the difficulty of math word problems. The human testers are more likely to struggle with questions that exhibit higher complexity. You should assign a higher difficulty score to the more challenging question.

When evaluating difficulty, consider the following factors derived from both computational complexity and linguistic analysis:

1. **Subtlety of Reasoning Required**: Questions that rely on nuanced implications or require multiple steps of inference to arrive at the correct answer tend to be harder. Look for questions where the link between the context and the answer is less direct or obvious.
2. **Breadth of Knowledge Needed**: Questions that pull in background knowledge spanning a wider range of concepts and situations lean toward being more difficult. Favor questions that integrate multiple strands of commonsense reasoning.
3. **Presence of Potentially Distracting or Misleading Elements**: Questions that include information that could point to the incorrect answer without careful scrutiny are often harder. The presence of such distractors requires a closer reading to arrive at the right answer.
4. **Avoidance of Obvious Associative Cues**: Easier questions sometimes include words or phrases that are strongly associated with one of the answers. More challenging questions tend to avoid such direct cues in favor of language that more subtly guides to the correct response.
5. **Degree of Answer Ambiguity**: In some cases, both answer options may seem plausible at first glance, with the correct answer determined by a crucial detail or distinction in the question. Lean toward these questions over ones where the answer is more immediately apparent.

In your assessment, you will be provided with both the questions their multiple-choice options, and the correct answer for each. Use the question and its correct choice to verify if the problem aligns with the factors mentioned above.

The scoring for each question should range from 1 to 10, where a score above 5 indicates that the question aligns strongly with these complexity indicators, suggesting a higher difficulty. Conversely, a score below 5 suggests a question is relatively less complex and easier to solve.

Start by offering a brief comparative analysis of the two questions based on the above criteria. Then, present your scores in the format: "[[score1, score2]]", where "score1" represents your assigned difficulty score for Question A, and "score2" stands for Question B. Ensure to maintain objectivity, eliminating any positional or length biases in your evaluation.

[The Start of Question A]

```
{question0}
```

[The End of Question A]

[The Start of Question A's Choices]

```
{choices0}
```

[The End of Question A's Choices]

[The Start of Question A's Correct Choice]

```
{target0}
```

[The End of Question A's Correct Choice]

[The Start of Question A]

{question1}

[The End of Question A]

[The Start of Question A's Choices]

{choices1}

[The End of Question A's Choices]

[The Start of Question A's Correct Choice]

{target1}

[The End of Question B's Correct Choice]

## J.3    Prompt templates for Benchmarking Performance

### E2H-AMC

Please take your time to thoroughly analyze and solve the following high-school math competition problem step by step. Your approach should be detailed, ensuring that each step of your reasoning is clearly explained to minimize errors and maximize understanding.

[PROBLEM_START]

{problem}

[PROBLEM_END]

While solving, consider all possible scenarios and subtleties involved in the problem. Each step should build upon the previous one logically, leading to a cohesive solution.

Once you arrive at the solution, please present the final answer enclosed in '☐'. Ensure the answer is displayed using appropriate LaTeX formatting to maintain mathematical precision and clarity.

### E2H-Codeforces

Please generate executable Python 3.10 code that directly solves the problem described below. The code should be ready to run without any modifications or additional comments. It must strictly follow Python 3.10 syntax and be formatted correctly for direct execution. Do not include explanations or comments within the code.

[PROBLEM_MAIN_START]

{problem_main}

[PROBLEM_MAIN_END]

[PROBLEM_NOTE_START]

{problem_note}

[PROBLEM_NOTE_END]

[INPUT_SPEC_START]

{input_spec}

[INPUT_SPEC_END]

[OUTPUT_SPEC_START]

```
{output_spec}
```
[OUTPUT_SPEC_END]

[SAMPLE_INPUTS_START]

```
{sample_inputs}
```
[SAMPLE_INPUTS_END]

[SAMPLE_OUTPUTS_START]

```
{sample_outputs}
```
[SAMPLE_OUTPUTS_END]

1. Please make sure to include correct import statements for any Python packages required by the solution at the start of the script.
2. When handling input within the code, utilize 'sys.stdin.readline()' instead of the 'input()' function.
3. The code should begin with ""'python" and conclude with "'''". Everything between these markers must be Python 3.10 code that is ready to execute as is. This code should be directly savable as a *.py file and fully functional to address the specified problem when run.

## E2H-Lichess

Analyze the chess position given in Forsyth-Edwards Notation (FEN) and determine the best possible next move for the side to move, with a focus on achieving a checkmate in one move. This chess puzzle, known as "mate in one," requires precise analysis. To guide your analysis, utilize the following resources:

1. Portable Game Notation (PGN): Helps understand the moves played so far.
2. Annotated PGN: Provides insights from the Stockfish chess engine, offering evaluations and annotations for strategic considerations that led to the current position.
3. FEN: Represents the current board setup accurately, showing which side is to move.
4. UCI Sequences: Use these to understand the sequence of moves leading up to the current position.

Your task is to find the objectively best move that results in checkmate in one turn. Present your answer in both Portable Game Notation (PGN) and Universal Chess Interface (UCI) formats. Analyze all candidate moves, explaining concretely why the selected move achieves checkmate, supported by engine evaluations and the current board position.

[PGN_START]

```
{pgn}
```
[PGN_END]

[ANNOTATED_PGN_START]

```
{annotated_pgn}
```
[ANNOTATED_PGN_END]

[FEN_START]

```
{fen}
```
[FEN_END]

[UCI_SEQUENCE_START]

```
{uci_seq}
```

[UCI_SEQUENCE_END]

Please reply with the predicted best next move in both PGN and UCI formats.

