# OpenReview forum: "Easy2Hard-Bench: Standardized Difficulty Labels for Profiling LLM Performance and Generalization"
_NeurIPS.cc/2024/Datasets_and_Benchmarks_Track — NeurIPS 2024 Track Datasets and Benchmarks Poster_

### Official Review · Reviewer_WtDm · 2024-07-27
**Refer to review**

**Rating:** 8
**Confidence:** 4
**Correctness:** Yes
**Clarity:** Yes

**Review:**

Strengths:
1. The concept of providing a difficulty score to measure LLM generalization is highly novel and intriguing. Evaluating generalization ability is crucial for the development and practical application of LLMs, and the diverse datasets from Easy2Hard offers a good perspective for further exploration. Moreover, the datasets cover a wide range of domains and difficult levels.

2.Comprehensive methodology is considered for difficulty estimation. IRT and Glicko-2 models for difficulty estimation is explored and provides moderate performance compared with human evaluation.

3. Experiments with LLMs offers insight of the model generation and validates the usage of Easy2Hard benchmark. The extensive experiments offer valuable insights into the performance and generalization capabilities of various LLMs, which can guide future research and development.

4. This paper is well-motivated and easy to follow.


Weaknesses:
1. The difficulty estimation scheme can be improved. For instance, the alignment with human evaluation on the ARC and Winogrande datasets are around 73%, and the overall alignment could be enhanced to better reflect human assessments. (If the authors can give a positive response to this point, I can adjust my rating score.)

2. The figure presentation of this paper can be improved.

**Strengths:**

Refer to review

**Additional Feedback:**

Refer to reivew

**Documentation:**

Yes

**Limitations:**

Yes

**Opportunities For Improvement:**

Refer to weakness.

**Relation To Prior Work:**

Yes

**Summary And Contributions:**

In this paper, the authors introduce Easy2Hard, a benchmark designed to advance research in LLMs generalization and understanding. Easy2Hard consists of six datasets spanning various domains, including mathematics and chess puzzles. A key contribution of this benchmark is the inclusion of a difficulty score, which quantifies the progression from easy to hard problems. Additionally, the authors conduct experiments using open-source LLMs, providing valuable insights into the generalization performance of current models.

---

> ### Author Rebuttal · Authors · 2024-08-17
>
> > The difficulty estimation scheme can be improved. For instance, the alignment with human evaluation on the ARC and Winogrande datasets are around 73%, and the overall alignment could be enhanced to better reflect human assessments. (If the authors can give a positive response to this point, I can adjust my rating score.)
>
> We appreciate the reviewer's detailed feedback. Indeed, as reported in Table 3, the matching accuracy between estimated difficulty and human rankings on ARC and Winogrande datasets is lower than on GSM8K. To better understand this observation, we dive deeper into the nature of these two datasets. Compared with GSM8K, the problems from these two datasets are more likely to show similar difficulties. Here are some problem pairs used in the human evaluation, where the participants are asked to choose which one is more difficult.
>
> **From the ARC dataset**
>
> * Problem 1: How are the stem of a tree and the stem of a flower most similar?
> (1) Both are soft.   (2) Both have thorns.   (3) Both support the plant.   (4) Both have woody bark.
> *2 participants chose it as the difficult problem, IRT difficulty: 0.787*
> * Problem 2: Which object has the greatest flexibility?
> (1) wooden block   (2) paper cup   (3) metal nail   (4) glass bottle
> *3 participants chose it as the difficult problem, IRT difficulty: 0.243*
>
> **From the Winogrande dataset**
>
> * Problem 1: Michael was on a diet, but cheated by eating junk food with Christopher.
> (1) Michael felt bad for ruining his plan.   (2) Christopher felt bad for ruining his plan.
> *3 participants chose it as the difficult problem, IRT difficulty: 0.090*
> * Problem 2: The game of chess was easy to play for Angela but not Rebecca because
> (1) Angela had an analytical mind.   (2) Rebecca had an analytical mind.
> *2 participants chose it as the difficult problem, IRT difficulty: 0.766*
>
> From these example pairs, we can see a large portion of problems in ARC and Winogrande datasets as closed and indistinguishable difficulties. A few participants of our human evaluation also specifically commented that they found ARC and Winogrande are hard to compare. We also quote their original comment below.
>
> * "To me, it’s hard to compare the question pairs in the second and third tasks (i.e., ARC and Winogrande datasets)."
> * "The last set (i.e., Wingrande dataset) was difficult to compare the pairs, since the difficulty of both options were quite similar."
>
> To further verify that the lower matching accuracy is caused by the closed difficulty distribution of the two datasets, we first calculate the proportion of problem pairs with low human confidence (2 vs. 3, or 3 vs. 2 votes, where there are 5 participants per problem). Low human confidence means that the number of participants on both sides are close, indicating that the problem difficulties are more indistinguihable to the humans. The results are shown below.
> |   |GSM8K|ARC|Winogrande|
> |---|-----|-----|-----|
> |Proportion of Pairs with Low Human Confidence |0.07|0.26|0.32|
>
> We see there are many more pairs of problems in ARC and Winogrande that have similar difficulties.
>
> As the next step, we try to exclude those pairs with low human confidence and recaculate the matchin accuracies and discrepancies to human rankings on the three datasets. The new results (and original ones in Table 3) are following.
>
> |   |Metric|GSM8K|ARC|Winogrande|
> |---|---|-----|-----|-----|
> |Original|Matching Acc.|0.942 $\pm$ 0.046|0.737 $\pm$ 0.047|0.734 $\pm$ 0.040|
> |Original|Avg. Per-pair Discrepancy|0.026 $\pm$ 0.015|0.096 $\pm$ 0.020|0.113 $\pm$ 0.027|
> |Excluded|Matching Acc.|0.963 $\pm$ 0.038|0.873 $\pm$ 0.112|0.847 $\pm$ 0.036|
> |Excluded|Avg. Per-pair Discrepancy|0.017 $\pm$ 0.012|0.041 $\pm$ 0.013|0.060 $\pm$ 0.018|
>
> We find that the matching accuracies increase around 10%-15% if we exclude the pairs with low human confidence. This further indicates that the main reason of lower matching accuracy on ARC and Winogrande is the larger proportion of problem pairs with indistinguishble difficulty to humans.
>
> > The figure presentation of this paper can be improved.
>
> Thank you for this detailed suggestion and kind reminder. We have reproduced the vector-formatted figures and we promise to replace all figures with the vector formatted ones in the next update of the manuscript.

---

> > ### Comment · Reviewer_WtDm · 2024-08-18
> > **Comment from Reviewer**
> >
> > Thank the authors for their response. The corresponding response has addressed my concerns, and I will revise my score.

---

### Official Review · Reviewer_2GjG · 2024-07-28

**Rating:** 7
**Confidence:** 4
**Correctness:** Yes
**Clarity:** Yes

**Review:**

This paper provides an important benchmark with finegrained difficulty level evaluation based on human evaluation. Careful difficulty estimation is conducted with human verification. However, I wonder why such "subjective" difficulty measurement is used to quantify difficulty; several papers have adopted more well-defined and "objective" computational complexity or "length" of algorithmic question to define difficulty of datapoints [1, 2, 3] and I wonder why these are not adopted at the beginning. Please provide more motivation and explanation and add more discussion with relevant benchmarks.


[1] Nphardeval: Dynamic benchmark on reasoning ability of large language models via complexity classes
[2] NPHardEval4V: A Dynamic Reasoning Benchmark of Multimodal Large Language Models
[3] DyVal: Dynamic Evaluation of Large Language Models for Reasoning Tasks

**Strengths:**

1. useful and timely benchmark with careful difficulty estimation
2. extensive experiments

**Additional Feedback:**

N/A

**Documentation:**

Yes

**Limitations:**

Yes

**Opportunities For Improvement:**

more motivation about the design/choice of difficulty level

**Relation To Prior Work:**

Need more discussion

**Summary And Contributions:**

This paper presents a benchmark with a refined, continuous valuation of difficulty, enabling a granular assessment of LLM capabilities across a spectrum of complexity. It also contains a diverse range of domains and tasks. It also includes human-verified difficulty assessments ensures the challenges reflect genuine cognitive hurdles, adding a layer of realism and relevance

---

> ### Author Rebuttal · Authors · 2024-08-17
>
> > This paper provides an important benchmark with finegrained difficulty level evaluation based on human evaluation. Careful difficulty estimation is conducted with human verification. However, I wonder why such "subjective" difficulty measurement is used to quantify difficulty; several papers have adopted more well-defined and "objective" computational complexity or "length" of algorithmic question to define difficulty of datapoints [1, 2, 3] and I wonder why these are not adopted at the beginning. Please provide more motivation and explanation and add more discussion with relevant benchmarks.
>
> We appreciate the reviewer for the thoughtful comment and for providing these related works about difficulty estimation.
>
> We would like to slightly revise the terms "subjective/objective" used by the reviewer to "complexity/statistics"-based in order to avoid unnecessary confusion. The difficulty estimation approach in [1,2,3] is based on computational complexity shared universally by the problems in the dataset. We also find some other works, such as [4,5], that also use specific properties shared by all problems to define the difficulty. Although our approach is based on human/model performance statistics, we think it should not be considered as "subjective" since it is based on the performance of a large crowd. Therefore, we call our approach statistics-based to distinguish from the approach of [1-5], which we call complexity-based.
>
> Comparing these two approches, we believe both of them are valuable and fruitful directions. Both approachs may have some limitations in practice.
>  - **Complexity-based** approach relies on deep and clear understanding of the problems' structure and nature. Many natural lauguage questions, although require reasoning ability, like ARC and Winogrande, does not enjoy this property. Therefore complexity-based approach is often limited to specific types of problems like algorithm questions.
>  - **Statistics-based** approach relies on high quality performance data of a large crowd. This can be expensive or even inaccessible.
>
> For our Easy2Hard-bench aiming at profiling LLM with difficulty attributes, we aim to include many popular types problems like chess and resoning problems that are not limitied to certain types of formal framework. Therefore we focus on the statistics-based approach.
>
> We completely agree that the complexity-based approach are also a valuable direction. We will discuss the papers mentioned and add to the related work section in the next update of manuscript. Thank you again for letting us know about these work.
>
> #### Reference
> - [1] Fan, Lizhou, et al. "Nphardeval: Dynamic benchmark on reasoning ability of large language models via complexity classes." arXiv preprint arXiv:2312.14890 (2023).
> - [2] Fan, Lizhou, et al. "NPHardEval4V: A Dynamic Reasoning Benchmark of Multimodal Large Language Models." arXiv preprint arXiv:2403.01777 (2024).
> - [3] Zhu, Kaijie, et al. "Dyval: Dynamic evaluation of large language models for reasoning tasks." The Twelfth International Conference on Learning Representations. 2024.
> - [4] Shoham, Ofir Ben, and Nadav Rappoport. "MedConceptsQA--Open Source Medical Concepts QA Benchmark." arXiv preprint arXiv:2405.07348 (2024).
> - [5] Bean, Andrew M., et al. "LINGOLY: A Benchmark of Olympiad-Level Linguistic Reasoning Puzzles in Low-Resource and Extinct Languages." arXiv preprint arXiv:2406.06196 (2024).

---

> ### Author Response · Authors · 2024-08-26
> **Reminder of Rebuttal Response**
>
> Dear Reviewer 2GjG,
>
> We sincerely appreciate the time and effort you have dedicated to reviewing our paper. As the discussion phase draws to a close, we have not yet received a response to our rebuttal. We hope that our rebuttal has addressed the concerns and clarifications you raised, and we would greatly value any additional feedback you may have.
>
> Your insights would be invaluable in helping us further refine and improve our work. Please share with us if there are any other questions or comments.
>
> Thank you once again for your thoughtful review.
>
> Best regards,
>
> The Authors of Paper 1405

---

### Official Review · Reviewer_H1Vy · 2024-07-28
**Useful, but for how long?**

**Rating:** 7
**Confidence:** 3
**Correctness:** As far as I can tell, no issues here.
**Clarity:** yes.

**Review:**

Quality: the research seems to be well done.
Clarity: the paper is clearly written and the figures are clear and easy to follow
Originality: the tasks aren't exactly new, but the difficulty ratings are. It's not a new idea, but as per the paper's claims, they are the first to offer a comprehensive benchmark covering multiple task types at a range of labelled difficulties.
Significance: while it is definitely useful to see how LLMs do on benchmarks by prompt difficulty, not much was said about whether this benchmark will have longevity. These days as soon as a benchmark is published it enters the training data for new generation LLMs and becomes obsolete. I wonder why we should believe that this benchmark should be expected to do better. That exactly the newly curated tasks were ones where LLMs had the most trouble suggests that perhaps they just aren't in the training data yet.

**Strengths:**

The basic concept seems useful for evaluating LLMs. More useful than a lot of the new benchmarks out there.

**Additional Feedback:**

no further comments.

**Documentation:**

yes.

**Ethics:**

no issues.

**Limitations:**

No issues here.

**Opportunities For Improvement:**

One question that wasn't answered but may be discussed in Appendix F is whether the tasks that are easy or hard for humans and for LLMs should be expected to be the same. I'd say no, but the paper assumes that they ought to be.
Also, I'd like to see more justification for why the particular tasks chosen should be considered representative. Chess is one that we know LLMs can be bad at, but I thought we'd moved beyond thinking of it as the gold standard for "intelligence". What was the rationale for choosing it as a task (as well as the others)? Easy availability of questions with human answers seems like the likely answer, but it's not a very satisfying one.

**Relation To Prior Work:**

yes.

**Summary And Contributions:**

Three newly curated tasks and 3 previously used tasks for evaluating LLM performance are organized into difficulty levels using both performance on LLMs and human performance as metrics. Both open source and proprietary SOTA LLMs are tested on the new benchmark tasks, validating the difficulty ratings.

---

> ### Author Rebuttal · Authors · 2024-08-17
>
> > Significance: while it is definitely useful to see how LLMs do on benchmarks by prompt difficulty, not much was said about whether this benchmark will have longevity. These days as soon as a benchmark is published it enters the training data for new generation LLMs and becomes obsolete. I wonder why we should believe that this benchmark should be expected to do better. That exactly the newly curated tasks were ones where LLMs had the most trouble suggests that perhaps they just aren't in the training data yet.
>
> We appreciate your thoughtful comment on the longevity of our benchmark. We understand the concern about benchmarks quickly becoming obsolete due to inclusion in training data for new LLMs. However, our difficulty estimation framework and many of our data sources (AMC, Lichess, Codeforces) are designed to be updatable. New problems can be easily added to the benchmark dataset, which are likely not seen by the new LLMs. We can update the datasets to include the latest problems and add timestamps to the problems in the datasets, ensuring that we can track when problems were introduced.
>
> It's worth noting that data leakage on our difficulty-profiling dataset may lead to more complex behavior than a simple accuracy or performance increase. This is because our evaluation outcomes are high-dimensional, which may make the benchmark more robust under leakage. Examples of this complexity can be seen in Figure 5 (a), (e), and (f) of our paper.
>
> Furthermore, for datasets like AMC, Codeforces, and Lichess, the data sources are annual competitions or game records from online servers. As long as these competitions continue or the servers remain active, new problems will be generated. These new problems have two key advantages:
>  - They come with clear time information, guaranteeing that they are not included in previous training data.
>  - Using the same estimation methods (such as IRT or Glicko-2), we can estimate the difficulty of new problems and compare them with the original problems.
>
> Additionally, we provide time information for each problem as tags. This enables users to filter out problems not included in the training data from the expanded dataset while evaluating LLMs, further enhancing the benchmark's longevity and relevance.
>
>
> > One question that wasn't answered but may be discussed in Appendix F is whether the tasks that are easy or hard for humans and for LLMs should be expected to be the same. I'd say no, but the paper assumes that they ought to be.
>
> We appreciate your insightful question regarding the alignment of task difficulty between humans and LLMs. We agree that they are not necessarily the same, which is reflected in Table 3 of our paper. The results show that while human and LLM difficulty ratings are well-aligned, they are not perfectly correlated. It is precisely because of this potential discrepancy that we conducted human evaluations to verify the degree of alignment.
>
> Our goal is to match or closely approximate human understanding of difficulty. The alignment we observed is not a given or assumed outcome. Although the performance data are sourced from LLM leaderboards, the resulting difficulty estimation shows high agreement with human evaluation. This alignment is achieved through two important steps in our methodology: outlier removal and model selection (details can be found in the appendix). These steps are used to remove noisy LLM performance data and better fit to GPT-4's explanations, which leads to better alignment with human understanding of difficulty, as shown in Table 3.
>
> > Also, I'd like to see more justification for why the particular tasks chosen should be considered representative. Chess is one that we know LLMs can be bad at, but I thought we'd moved beyond thinking of it as the gold standard for "intelligence". What was the rationale for choosing it as a task (as well as the others)? Easy availability of questions with human answers seems like the likely answer, but it's not a very satisfying one.
>
> We appreciate your thoughtful question about the rationale behind our task selection. It's important to clarify that our paper does not aim to answer the fundamental and long-debated question of what constitutes the gold standard for "intelligence." Instead, our goal is to capture domains and problems where there is a human-consensus understanding of difficulty and to evaluate LLM behavior within that context.
>
> To this end, we considered common domains such as math, coding, puzzles, and reasoning, which have well-recognized difficulty concepts. While these domains are not exhaustive, they cover several of the most common types of questions that have widely-accepted difficulty levels. Chess, in particular, was chosen not as a gold standard for intelligence, but as one of several tasks that fit our criteria of having a clear difficulty progression and established human benchmarks.
>
> Our selection process prioritized domains where difficulty could be objectively measured and compared across human and AI performance. The availability of questions with human answers was indeed a factor, but it was not the sole or primary consideration. Rather, we sought tasks that would provide meaningful insights into how LLMs handle problems of varying difficulties across different cognitive domains.
>
> We acknowledge that this selection is not complete or perfect, but it serves as a starting point for understanding LLM behavior across a range of difficulty levels in commonly recognized problem-solving domains. Future work could certainly expand on this selection to include a broader range of tasks and domains.

---

> ### Author Response · Authors · 2024-08-26
> **Reminder of Rebuttal Response**
>
> Dear Reviewer H1Vy,
>
> We sincerely appreciate the time and effort you have dedicated to reviewing our paper. As the discussion phase draws to a close, we have not yet received a response to our rebuttal. We hope that our rebuttal has addressed the concerns and clarifications you raised, and we would greatly value any additional feedback you may have.
>
> Your insights would be invaluable in helping us further refine and improve our work. Please share with us if there are any other questions or comments.
>
> Best regards,
>
> The Authors of Paper 1405

---

### Official Review · Reviewer_q7Pi · 2024-08-07

**Rating:** 7
**Confidence:** 4
**Clarity:** The paper is overall well written.

**Review:**

The paper is overall good with much engineering effort on dataset curation, automatic hardness quantification, human verification, benchmarking and more investigations on easy-to-hard generalization.

**Strengths:**

1. The paper is well motivated for studying the difficulty level issues -- and it is closely related to the broader research community on easy-to-hard problem generation and evaluation.
2. The paper is well-structured with detailed, sound and comprehensive description of each component in the pipeline. I appreciate the huge engineering efforts behind it.

**Additional Feedback:**

See improvements.

**Correctness:**

The claims are sound for having an evaluation dataset with varying levels of difficulty, The dataset and benchmark protocol is overall constructed in a sound way.

**Documentation:**

The dataset is still not open-sourced but the authors provide documentation and other materials in the supplementary. The documentation and supplementary materials are detailed and solid for those necessary information -- like maintenance, license, ethical issues, etc.

**Ethics:**

No.

**Limitations:**

The authors address the limitation in the Appendix.

**Opportunities For Improvement:**

1. I am not sure about the codeforce and AMC, but from my understanding of lichess puzzle dataset, the Glicko-2 ratio seems to be a common metric for it and the lichess puzzle dataset has already had this information -- the author should make it more clear with more clarifications and citations,  instead of claiming it as your own contribution (the authors use terms "innovative application" while I do not fully agree it is that innovative). In addition, more comparisons and analysis about why thes particular ways of modelling the difficulty work better can strengthen the motivation of applying these metrics.

2. The authors mention that they exclude the multi-step solution in the Lichess dataset -- however most puzzles in the dataset should be multi-step. This additional data preprocessing can somehow weaken the conclusion.

3. It is a bit weird for the quantification of GSM8K, ARC, and Winogrande difficulty with LLM's performance on the open-llm leader board. So you are  benchmarking the performance on different difficulty level while such difficulty level is also extracted by the LLM performance (like the chicken and egg problem). But to be honest I am not sure about potential better way to address this. Do the authors have any thoughts on this issue?

**Relation To Prior Work:**

The related work is clearly addressed.

**Summary And Contributions:**

The paper proposed Easy2Hard-Bench, which tries to standardized difficulty labels for different language tasks.

The paper provides:
[1] A dataset consisting of diverse domain of language tasks and a framework to quantify the difficulty for different domains.
[2] Benchmarking several advanced LLMs on the curated dataset for their peroformance on easy, medium, hard problems.
[3] Further analysis on easy2hard generalization.

---

> ### Author Rebuttal · Authors · 2024-08-17
>
> > I am not sure about the codeforce and AMC, but from my understanding of lichess puzzle dataset, the Glicko-2 ratio seems to be a common metric for it and the lichess puzzle dataset has already had this information -- the author should make it more clear with more clarifications and citations, instead of claiming it as your own contribution (the authors use terms "innovative application" while I do not fully agree it is that innovative).
>
> We appreciate your comment and suggestion. In the paper, when we say "innovative" application, as the reviewer mentioned, we do not refer to the Glicko-2 algorithm. Instead, we are referring to the way we model problem difficulty with the Glicko-2 framework: we treat each attempt on a problem as a discrete "game," and the problem itself can be thought of as one of the players. Indeed, as you mentioned, Lichess's open database already provides the problem difficulty calculated in this way (described in Appendix E.1), while for codeforces, we calculated from scratch. Therefore, we completely agree it is inappropriate to say this way of calculating problem difficulty is "innovative." We will revise this adjective and be careful of related descriptions in the manuscript to ensure our rigor and professionality.
>
> > In addition, more comparisons and analysis about why these particular ways of modelling the difficulty work better can strengthen the motivation of applying these metrics.
>
> Thank you for this thoughtful suggestion. To demonstrate the effectiveness and necessity of modeling frameworks like IRT, we conduct a sample ablation study on the GSM8K, ARC, and Winogrande datasets. In Table 3 in the manuscript, we show the matching accuracy between the IRT-based difficulty and human rankings. Now, we can also treat the negated LLMs' average accuracy on a problem as a measurement of problem difficulty. We show the matching accuracy between the negative LLM average accuracy and human rankings as well. All the other setups, like the set of LLMs considered, are not changed. The comparison results are shown in the following table.
>
> |   |GSM8K|ARC|Winogrande|
> |---|-----|-----|-----|
> |IRT-based Difficulty vs Human (matching acc.)|0.942 $\pm$ 0.046|0.737 $\pm$ 0.047|0.734 $\pm$ 0.040|
> |Negative LLM Average Acc. vs Human (matching acc.)|0.882 $\pm$ 0.020|0.711 $\pm$ 0.048|0.612 $\pm$ 0.009|
>
> We clearly see that the IRT-based difficulty aligns significantly better with human understanding. IRT takes the participants' ability as learning parameters and jointly models with problem difficulty. We think this allows IRT models difficulty better than naive average of performance of a problem, when the set of participants have very different abilities. We will also add more ablation studies, like similar desmonstration for Glicko-2 framework to the manusript.
>
> > The authors mention that they exclude the multi-step solution in the Lichess dataset – however most puzzles in the dataset should be multi-step. This additional data preprocessing can somehow weaken the conclusion.
>
> Thank you for the detailed comment. Focusing on chess puzzles with single-step solutions is actually not a relatively additional/unthoughtful decision but instead a trade-off between the completeness and accessibility of the benchmark. Multi-step chess puzzles require multiple rounds of interactions with LLM during evaluation. For example, when LLM predicts the next move, we need to check its correctness; if correct, we show the next move by the puzzle's side. This multi-round interaction/QA with LLM is more complex than the common one-time inference evaluation setup used by the other datasets in our benchmarks. Moreover, it requires much more computations to evaluate each problem. Given that (1) the one-step puzzles in Lichess are already abundant (71,763 train and 5,000 test) and (2) the community also focuses more on one-step chess puzzles (e.g., [1], [2]), we decide to not include multi-step puzzles for now, for the accessibility of the proposed benchmark. Multi-step chess puzzles can be a future work. Since the difficulty estimation method and process are the same, it only requires the implementation of a multi-round QA evaluation.
>
> > It is a bit weird for the quantification of GSM8K, ARC, and Winogrande difficulty with LLM's performance on the open-llm leader board. So you are benchmarking the performance on different difficulty level while such difficulty level is also extracted by the LLM performance (like the chicken and egg problem). But to be honest I am not sure about potential better way to address this. Do the authors have any thoughts on this issue?
>
> We appreciate your thoughtful question. We think this is not a chicken-and-egg problem.
>
> Although we are using LLMs' performance as data source to model problem difficulty, our goal is to simulate and align-well to the human's understanding. From Table 3, we see that the resulted difficulty estimation has high agreement with human's rankings.
>
> Moreover, we do not directly apply IRT models on the complete raw LLM performance data. Two important steps: outlier removal and model selection (detailed in Appendix E.1 and G.1) are applied to remove noisy LLM performance data, which lead to better alignment with human understanding of difficulty in Table 3.
>
> #### References
> - [1] Burns, Collin, et al. "Weak-to-strong generalization: Eliciting strong capabilities with weak supervision." arXiv preprint arXiv:2312.09390 (2023).
> - [2] Feng, Xidong, et al. "Chessgpt: Bridging policy learning and language modeling." Advances in Neural Information Processing Systems 36 (2024).

---

> ### Author Response · Authors · 2024-08-26
> **Reminder of Rebuttal Response**
>
> Dear Reviewer q7Pi,
>
> We sincerely appreciate the time and effort you have dedicated to reviewing our paper. As the discussion phase draws to a close, we have not yet received a response to our rebuttal. We hope that our rebuttal has addressed the concerns and clarifications you raised, and we would greatly value any additional feedback you may have.
>
> Your insights would be invaluable in helping us further refine and improve our work. Please share with us if there are any other questions or comments.
>
> Best regards,
>
> The Authors of Paper 1405

---

### Decision · Program_Chairs · 2024-09-26

**Decision:**

Accept (Poster)

**Comment:**

**Summary of Contributions**

- The paper proposed Easy2Hard-Bench, which tries to standardized difficulty labels for different language tasks.

**Strengths**

- The paper is well motivated for studying the difficulty level issues -- and it is closely related to the broader research community on easy-to-hard problem generation and evaluation.
- It contains a diverse range of domains and tasks.
- It includes human-verified difficulty assessments ensures the challenges reflect genuine cognitive hurdles, adding a layer of realism and relevance.

**Weaknesses**

- The difficulty estimation scheme can be improved. For instance, the alignment with human evaluation on the ARC and Winogrande datasets are around 73%, and the overall alignment could be enhanced to better reflect human assessments.
- The presentation of figures can be improved.

**Discussion of Reviewers' Opinions**

- Since all reviewers give positive scores, no further discussion is given.

**Final Recommendation**

- After carefully considering the reviews and my own evaluation, I recommend accept this paper. The concerns are valid and should be addressed in the final version, but they do not detract from the overall contribution of the work.